# ENCRYPTION-FRIENDLY LLM ARCHITECTURE

**Donghwan Rho**[1*]**, Taeseong Kim**[1*]**, Minje Park**[2]**, Jung Woo Kim**[2]**, Hyunsik Chae**[1]**,
Ernest K. Ryu**[3†]**, Jung Hee Cheon**[12†]

[1]Seoul National University, Department of Mathematical Sciences
{ dongwhan_rho, kts1023, hsc1403, jhcheon }@snu.ac.kr
[2]CryptoLab Inc,
{ minje, jungwoo.kim }@cryptolab.co.kr
[3]UCLA, Department of Mathematics
eryu@math.ucla.edu

## ABSTRACT

Large language models (LLMs) offer personalized responses based on user interactions, but this use case raises serious privacy concerns. Homomorphic encryption (HE) is a cryptographic protocol supporting arithmetic computations in encrypted states and provides a potential solution for privacy-preserving machine learning (PPML). However, the computational intensity of transformers poses challenges for applying HE to LLMs. In this work, we propose a modified HE-friendly transformer architecture with an emphasis on inference following personalized (private) fine-tuning. Utilizing LoRA fine-tuning and Gaussian kernels, we achieve significant computational speedups—$6.94\times$ for fine-tuning and $2.3\times$ for inference—while maintaining performance comparable to plaintext models. Our findings provide a viable proof of concept for offering privacy-preserving LLM services in areas where data protection is crucial. Our code is available on GitHub[1].

## 1 INTRODUCTION

The advent of large language models (LLMs) such as the BERT series (Devlin et al., 2019; Liu et al., 2019; Sanh et al., 2019; Lan et al., 2020; Clark et al., 2020; He et al., 2021), the GPT series (Radford, 2018; Radford et al., 2019; Tom B. Brown et al., 2020; OpenAI, 2023), and ChatGPT (OpenAI, 2024) kick-started a new era of natural language processing (NLP) and artificial intelligence (AI). One of the many capabilities of LLMs that has received much attention is their ability to provide personalized responses based on user interactions, especially with the use of fine-tuning. However, this use case raises serious concerns about user privacy. In response, regulations such as the GDPR (European Union, 2016) and CCPA (State of California, 2018) have been amended. In Italy, ChatGPT was even temporarily banned (McCallum, 2023), and several major corporations, including Apple and Samsung, have restricted its use within their companies (Mok, 2023).

Privacy-preserving machine learning (PPML) refers to methods that use machine learning while protecting data privacy. Techniques for PPML include secure multi-party computation (MPC) (Yao, 1982), differential privacy (Dwork, 2006), and homomorphic encryption (HE) (Rivest et al., 1978; Gentry, 2009). Among these, only MPC and HE offer provable security based on cryptographic assumptions. MPC utilizes communications between parties, but these communications can make it challenging to accelerate and parallelize the heavy computation of neural networks. In contrast, HE supports arithmetic computations in encrypted states without requiring communications. Since the pioneering work of Gentry (2009), several HE schemes have been developed (Brakerski, 2012; Brakerski et al., 2014; Ducas & Micciancio, 2015; Chillotti et al., 2016; Cheon et al., 2017). Notably, the CKKS (Cheon et al., 2017) scheme is particularly efficient for evaluating large-scale real-valued (as opposed to integer-valued) data in parallel and is widely used in the PPML literature (Han et al., 2019; Lee et al., 2022a;b;c; 2023b).

---

[*]Equal contribution. Order determined by a coin toss
[†]Co-senior authors.
[1]https://github.com/Donghwan-Rho/Encryption-friendly_LLM_Architecture

In theory, homomorphic encryption (HE) presents an elegant solution to the privacy concerns associated with LLMs. However, despite the significant recent progress in the theory and implementation of HE operations, protecting LLMs with HE remains a challenge due to their computational scale. Transformer models (Vaswani et al., 2017) famously rely on numerous matrix multiplications and various non-polynomial operations, and directly adapting these operations to HE results in significant computation time and precision loss.

In this work, we propose a modified HE-friendly transformer architecture with an emphasis on inference following personalized (private) fine-tuning. We point out that prior work on homomorphically encrypted transformers often overlooked fine-tuning due to its complexity. Our approach has two main algorithmic components: LoRA (Hu et al., 2022) fine-tuning and the replacement of softmax in attention with Gaussian kernels (GK). We show that LoRA and GK significantly accelerate the encrypted transformer computation under HE while maintaining performance levels comparable to those of the plaintext model. Experimental results of our modified BERT model on encrypted data using the CKKS scheme demonstrate its ability to *securely* process natural language data. Our findings show promise for offering privacy-preserving LLM services in areas where data protection is crucial.

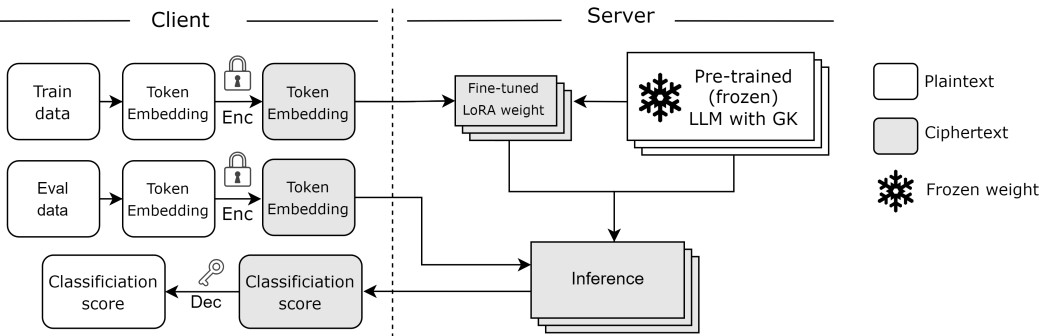

Figure 1: Proposed privacy-preserving LLM under homomorphic encryption (HE). HE cryptographically protects user's fine-tuning and inference data. We resolve two computational bottlenecks. First, we reduce the size of ciphertext-ciphertext matrix multiplication (CCMM) using LoRA fine-tuning. Second, we avoid the softmax computation, which is notoriously challenging to compute under HE, and replace it with a much simpler Gaussian kernel (GK).

## 1.1 PRIOR WORK

**Transformer-based language models and LoRA.** Since the advent of attention (Vaswani et al., 2017), the transformer has become the standard of language models. There are three types of transformer-based language models. Encoder-only models, including BERT series (Devlin et al., 2019; Liu et al., 2019; Sanh et al., 2019; Lan et al., 2020; Clark et al., 2020; He et al., 2021) output embeddings for inputs that can be used for downstream tasks. Encoder-decoder models, which use the original transformer architecture, such as MarianMT (Junczys-Dowmunt et al., 2019), T5 (Raffel et al., 2020), BART (Lewis et al., 2020), mBART (Liu et al., 2020), and mT5 (Xue et al., 2021), are used for translation, summarizing, etc. Decoder-only models, including GPT series (Radford, 2018; Radford et al., 2019; Tom B. Brown et al., 2020; OpenAI, 2023) and Llama series (Touvron et al., 2023a;b; Dubey et al., 2024), generate sentences to the user's query. These large language models (LLMs) follow the scaling law (Kaplan et al., 2020), so the scale of LLMs tends to increase more and more. Hence, these models require a huge amount of memory capacity for inference and fine-tuning. To overcome this issue, LoRA (Hu et al., 2022) is mainly used to fine-tune a pre-trained LLM. Freezing all other weights, LoRA adapters are added to important layers of the model, such as attention layers during fine-tuning. Using LoRA, one can fine-tune LLMs, updating only less than 1% of all of the parameters.

**Privacy-preserving transformer using HE.** Many researchers have explored privacy-preserving algorithms leveraging HE for transformer models. There are two main scenarios: interactive, which

combines secure MPC with HE, and non-interactive, which relies only on HE. In interactive scenarios, encrypted computation time can be reduced through communications between parties. THE-X (Chen et al., 2022) proposed the first protocol for BERT-tiny inference, introducing HE-friendly workflows and non-polynomial evaluations via communications. Subsequent research (Hao et al., 2022; Li et al., 2023; Akimoto et al., 2023; Dong et al., 2023; Pang et al., 2024) enhanced computation time and reduced communication costs.

However, interactive approaches may struggle with large-scale communication as model size increases, and they also require data owners to be online during computation. To address these, non-interactive research is being conducted on the other side. Zimerman et al. (2024) introduced the first HE-friendly transformer model, replacing the original transformer with a HE-friendly structure, minimizing the approximation domain, and obtaining pre-trained weight with the changed structure. Using these weights, they performed inference with BERT non-interactively. NEXUS (Zhang et al., 2024a) further proposed a non-interactive BERT inference method without re-training, using polynomial approximations for non-polynomial operations. More recently, Park et al. (2024) proposed Powerformer, which, like our method, proposed replacing softmax in the attention with their BRP-max function for homomorphic inference.

Fine-tuning plays a crucial role in improving transformer models and providing more personalized responses. However, previous works have primarily focused on secure inference, largely avoiding the challenge of fine-tuning due to the significant computational complexity involved, especially in non-interactive settings. Only a few attempts, such as those by Lee et al. (2022b) and HETAL (Lee et al., 2023b), have explored fine-tuning, focusing exclusively on the classification head while leaving other key components like attention layers and feed-forward networks untouched. P3EFT (Li et al., 2024) also addresses fine-tuning of foundation models, but it concentrates more on energy consumption during fine-tuning.

## 1.2 CONTRIBUTIONS

We propose a homomorphic encryption (HE) friendly transformer architecture with an emphasis on inference following personalized (private) fine-tuning. We resolve two computational bottlenecks of the HE transformer model: (i) using LoRA, we avoid large ciphertext-ciphertext matrix multiplications (CCMMs), and (ii) we use a simpler Gaussian kernel (GK) to replace the softmax layer, which is notoriously challenging to compute under HE. Experiments on an HE-encrypted BERT-style transformer demonstrate a speedup of $6.94\times$ for fine-tuning and $2.3\times$ for inference.

## 2 SERVER-CLIENT COMPUTATION MODEL AND PRELIMINARIES

In this section, we first state the server-client computation model, referred to as the *threat model* in the cryptography community. We then quickly review the relevant basic concepts and introduce the notation for homomorphic encryption and large language models.

### 2.1 SERVER-CLIENT COMPUTATION MODEL

Our server-client computation model (threat model) involves a client with private data, outsourcing fine-tuning and inference to the server, an LLM service provider. See Figure 1. Specifically, the client sends encrypted token-embedded data to the server, and the server fine-tunes the model, generating customized encrypted LoRA weights. Subsequently, the server performs inference using encrypted inference data, plaintext pre-trained weights, and encrypted LoRA weights. The server returns the encrypted inference results to the client. The token embedding layer weights are not encrypted and are not updated throughout fine-tuning.

Our model is based on the semi-honest security model. The adversary adheres to the protocol but is allowed to collect all outputs from the model. Since the client's input data is encrypted with the CKKS ciphertext, the entire security model relies on the semantic security of the underlying CKKS.

The user data used throughout fine-tuning and inference is cryptographically protected, even from the LLM service provider. Both the pre-trained weights of the LLM and the fine-tuned LoRA weights reside on the server and are not shared with the user. However, we clarify that this does not mean the

LLM weights are protected in the strict cryptographic sense, and our proposed approach does not address the model weight stealing problem (Tramèr et al., 2016; Carlini et al., 2024).

## 2.2 HOMOMORPHIC ENCRYPTION AND CKKS

Homomorphic encryption (HE) is an encryption method that allows computations to be performed in an encrypted state without decryption. CKKS (Cheon et al., 2017) is one of the HE schemes that allows one to encrypt real or complex data as a polynomial and perform approximate arithmetic on the encrypted data. By packing multiple real or complex data into a ciphertext, CKKS supports single instruction multiple data (SIMD) operations in plain/ciphertexts. Plaintext denotes an unencrypted polynomial containing a message vector in the context related to HE. For a more detailed description, refer to Appendix B or the original reference Cheon et al. (2017).

- **Addition** (Add)**:** Given two ciphertexts or plaintext and ciphertext, evaluate component-wise addition.
- **Multiplication** (Mult)**:** Given two ciphertexts or plaintext and ciphertext, evaluate component-wise multiplication. We express the plaintext-ciphertext component-wise multiplcation as pMult.
- **Rotation** (Rot)**:** Given a ciphertext ct encrypting message vector $(m_0, m_1, \ldots, m_{N/2-1})$, return $\text{ct}_{rot_i}$ encrypting message vector almost same with $(m_i, \ldots, m_{N/2-1}, m_0, \ldots, m_{i-1})$. We denote $\text{Rot}(\cdot, i)$ for $i$th left rotation. We express the same operation in plaintext as pRot.

Note that Mult and Rot require an additional key-switching operation. So, both take more computation time than other operations. For concrete time measurements, see Table 5 in Appendix E.

CKKS is one of the leveled HE schemes. A CKKS ciphertext has a restricted number of (p)Mult operations, so-called *level*, which is determined at the beginning of encryption. The modulus $Q$, associated with the level, is reduced after these operations. Once we exhaust the available level budget, we cannot proceed to the next (p)Mult operation. To recover the level budget, we can use *bootstrapping* (BTS) (Cheon et al., 2018). Since BTS is the most expensive HE operation, which is over $100\times$ slower than Mult, Rot, it is crucial to minimize its usage through effective designing of the algorithm. In fact, the time accounted for BTS constitutes a large portion of the overall computation time (Lee et al., 2022a; Zhang et al., 2024a).

**Matrix multiplications: PCMM and CCMM.** While evaluating the transformer model under HE, we use two kinds of homomorphic matrix multiplications: plaintext-ciphertext matrix multiplication (PCMM) and ciphertext-ciphertext matrix multiplication (CCMM). Homomprhic matrix multiplication requires numerous HE operations, such as (p)Mult, (p)Rot and Add, for each matrix, which are slower when operating on ciphertext compared to plaintext as shown later in Table 5.
As a result, *PCMM is much faster than CCMM*.

There have been several studies (Jiang et al., 2018; Jang et al., 2022; Rizomiliotis & Triakosia, 2022; Zheng et al., 2023; Bae et al., 2024) to make homomorphic matrix multiplication more efficient. In this paper, we follow the JKLS (Jiang et al., 2018) algorithm, which provides the fastest known HE computation algorithm when the matrix elements can be packed in a ciphertext. They also propose parallel homomorphic matrix multiplication. We use this parallel computation to evaluate large-size matrix multiplication, where the size is more than available ciphertext space, by repeating block-wise matrix multiplcation. Thus, the computation time gap between PCMM and CCMM will be proportional to the number of block matrices. For more details, refer to Appendix C.

**Polynomial approximations of non-polynomial functions.** Since HE can only support polynomial operations, all non-polynomial operations (such as division, max function, etc.) must be replaced by their polynomial approximations. The complexity of polynomial approximation highly depends on the input range and the nature of the function being approximated.

## 2.3 LARGE LANGUAGE MODELS, ATTENTION LAYERS, AND LORA FINE-TUNING

Standard large language models (LLMs), such as BERT (Devlin et al., 2019), GPT-4 (OpenAI, 2023), and Llama-3 (Dubey et al., 2024) are constructed by stacking many transformer layers. We explain

the standard attention mechanism in a transformer (Vaswani et al., 2017) layer. We do not explain the feed-forward network (FFN), another component of the transformer, as it has not been modified in our work. Suppose that $L$ and $n$ represent sequence length and embedding dimension, respectively. Given $Q, K, V \in \mathbb{R}^{L \times n}$, the standard attention is calculated as

$$\text{Attention}(Q, K, V) = \text{Softmax}\left(\frac{QK^\top}{\sqrt{n}}\right) V \in \mathbb{R}^{L \times n}. \tag{1}$$

The rows of $Q$, $K$, and $V$ are respectively referred to as query, key, and value vectors. Here, the softmax function (Bridle, 1989) is applied row-wise, normalizing the similarity scores between each query and key vectors into probability distributions, which serve to weigh the importance of each value vector. Later in Section 4, we specifically discuss why the softmax function is numerically challenging to compute under homomorphic encryption (HE).

Low-Rank Adaptation (LoRA) (Hu et al., 2022) of large language models has become the most widely used parameter-efficient fine-tuning scheme for LLMs. Given a weight matrix $W \in \mathbb{R}^{n \times n}$ of a linear layer, LoRA updates $W$ with an update $\Delta W = AB$, where $r \ll n$, $A \in \mathbb{R}^{n \times r}$, and $B \in \mathbb{R}^{r \times n}$, so that the linear layer's action becomes $X \mapsto X(W + AB)$ for an input $X \in \mathbb{R}^{L \times n}$. In this LoRA fine-tuning, the pre-trained weight $W$ is frozen while only $A$ and $B$ are trained.

## 3 SPEEDUP WITH LoRA: AVOIDING LARGE CCMM

**Bottleneck 1: Full fine-tuning incurs large CCMM.** Personalized fine-tuning data is transmitted to the server encrypted, and the fine-tuning updates, which depend on the users' private data, must also be stored encrypted. Subsequently, encrypted inference data is transmitted, and the transformer performs CCMM computations. Full fine-tuning updates all weights and, therefore, lead to many large CCMM calculations.

Specifically, consider a pre-trained linear layer with weight matrix $W \in \mathbb{R}^{n \times n}$ stored in plaintext. If we update $W$ to $W + \Delta W$ where $\Delta W \in \mathbb{R}^{n \times n}$, then evaluating the linear layer

$$X(W + \Delta W) = XW + X\Delta W, \qquad \text{for } X \in \mathbb{R}^{L \times n}$$

costs $\mathcal{O}(n^2)$ HE operations, where $\mathcal{O}(\cdot)$ only considers the dependence on $n$. However, while $XW$ can be evaluated with PCMM, $X\Delta W$ must be evaluated with the costly CCMM. This is because $W$ is not encrypted, but $\Delta W$ must be encrypted.

**Accelerating homomorphic matrix-multiplication with LoRA.** LoRA (Hu et al., 2022) fine-tuning alleviates the cost of CCMM by reducing the size of CCMMs. We clarify that the primary benefit of LoRA in the usual plaintext application is the reduced memory footprint due to having fewer optimizer memory states. Under HE, however, the main benefit is also converting large CCMMs into large PCMMs and reducing the computational cost.

Again, consider a pre-trained linear layer with weight matrix $W \in \mathbb{R}^{n \times n}$ stored in plaintext. However, consider LoRA fine-tuning updating $W$ to $W + AB$, where $r \ll n$, $A \in \mathbb{R}^{n \times r}$, and $B \in \mathbb{R}^{r \times n}$. Then, evaluating the linear layer

$$X(W + AB) = XW + (XA)B, \qquad \text{for } X \in \mathbb{R}^{L \times n}$$

requires an $\mathcal{O}(n^2)$-cost PCMM to evaluate $XW$ and two $\mathcal{O}(nr)$-cost CCMMs to evaluate $(XA)B$, where $\mathcal{O}(\cdot)$ only considers the dependence on $n$ and $r$. See Appendix C for further details. Since $A$ and $B$ are encrypted, evaluating $(XA)B$ still requires CCMMs, but the CCMMs are smaller than $X\Delta W$ by a factor of $n/r$. The difference in computational cost is significant, as shown in Table 1a and Appendix G.

Finally, we mention, but without describing the details, that an analogous consideration can be applied to backpropagation. (We remind the readers that we also perform fine-tuning under HE.)

**Reducing optimizer states and inverse square root with LoRA.** We fine-tune the transformer under HE using a variant of AdamW optimizer. The precise version of AdamW that we implement under HE is described in Appendix 5.2. For training and fine-tuning transformers, it is well known

that adaptive optimizers like AdamW significantly outperform non-adaptive optimizers such as SGD (Zhang et al., 2024b).

However, the inverse square root function mapping is $x \mapsto 1/\sqrt{x}$. This is also a non-polynomial function that tends to be difficult to use under HE. In general, due to wide input ranges for division, BTS is required for each ciphertext containing weights during evaluation. Given the large number of parameters in the transformer model, the optimizer step can create a computation time bottleneck. In fact, the optimizer's computation time exceeds that of the transformer block evaluation in our target model under full fine-tuning, as presented in Section 5.1.

LoRA alleviates this computational cost by substantially reducing the number of parameters being fine-tuned, thereby reducing the number of inverse square root operations needed in the AdamW optimizer. In our target 2-layer BERT model, full fine-tuning requires 368 ciphertexts containing weights, whereas LoRA only needs 15, offering a clear advantage in optimizer computation time.

## 4 SPEEDUP WITH GAUSSIAN KERNEL: POLY-APX-FRIENDLY DESIGN

**Bottleneck 2: Softmax is hard to evaluate under HE.** Since HE only supports polynomial operations, non-polynomial functions must be approximated as polynomials. Traditionally, the softmax function is evaluated with

$$\text{Softmax}\left(x_1, x_2, \cdots, x_n\right)_i = \frac{\exp\left(x_i - \alpha\right)}{\sum_j \exp\left(x_j - \alpha\right)}, \quad \text{where } \alpha = \max_{1 \leq j \leq n} \left\{x_j\right\}.$$

Subtracting $\alpha$ is crucial to avoid taking the exponential of a large positive number, which would lead to numerical instabilities. Under HE, however, the exponential, division, and max functions are non-polynomial functions that must be approximated. To clarify, the max function is conventionally evaluated using comparisons and if-statements, but these are non-polynomial functions that require polynomial approximations under HE.

While there is some recent work on efficiently approximating softmax function as polynomials (Badawi et al., 2020; Jin et al., 2020; Hong et al., 2022; Lee et al., 2022c; 2023b), softmax is still considered numerically challenging to evaluate under HE. In this section, we resolve this issue by avoiding the softmax function altogether and introducing the Gaussian kernel (GK) as an alternative.

**GK attention.** The standard attention layer (1) obtains the attention scores using the softmax function, which is difficult to approximate under HE. We can view the softmax as normalizing the outputs of a scaled exponential kernel, where "kernel" is used in the sense of RKHS kernel methods of classical machine learning (Berlinet & Thomas-Agnan, 2011). However, fundamentally, there is no requirement to use the exponential kernel, nor is there a necessity for normalizing the scores.

We propose the alternative *Gaussian kernel attention*:

$$\text{GK-Attention}(Q, K, V) = S(Q, K)V$$
$$S(Q, K)_{ij} = \exp\left(-\frac{1}{2\sqrt{n}} \left\|Q_{i,:} - K_{j,:}\right\|_2^2\right), \ i, j = 1, \ldots, L, \quad (2)$$

where $\|\cdot\|_2$ is the $L_2$ norm, $n$ is the hidden dimension, and $Q_{i,:}$ and $K_{j,:}$ mean the $i$th and $j$th row of $Q$ and $K$. Compared to the standard attention, the scaled exponential kernel is replaced with a Gaussian kernel and we do not perform normalization. The Gaussian kernel is much easier to evaluate than the softmax function for the following reasons. First, there is no need for division. (The factor $1/(2\sqrt{n})$ is fixed, so the reciprocal can be pre-computed and then multiplied.) Second, there is no need to compute the $\max$ function, which is difficult to approximate under HE. Third, $\exp(x)$ only needs to be approximated for $x \leq 0$, the region in which $\exp(x)$ does not blow up and therefore is much easier to approximate numerically. Specifically, we use

$$\exp(x) \approx p_k(x) := \left(1 + \frac{x}{2^k}\right)^{2^k} \text{ for } k \in \mathbb{N},$$

which is very accurate for the range $\left[-2^k, 0\right]$. See Appendix F for further discussions on the polynomial approximation of $\exp(x)$.

Finally, we point out that in contexts unrelated to encryption or privacy, the prior work of Richter & Wattenhofer (2020) made the observation that normalizations (divisions) are not necessary for the attention layers and that the prior work of Lu et al. (2021) and Chen et al. (2021) used Gaussian kernel in place of the softmax. Specifically, Richter & Wattenhofer (2020) removed normalizations to improve the theoretical and practical characteristics of transformers, and Lu et al. (2021) and Chen et al. (2021) further used a low-rank approximation to the Gaussian kernel to present a linear-complexity attention layer accommodating long context lengths.

## 5 EXPERIMENTAL RESULTS

### 5.1 EXPERIMENTAL SETUP

In this subsection, we quickly describe some core aspects of the experimental setup. Some details, such as the penalty training trick and hyperparameters, are deferred to Appendix E.

**Implementation of homomorphic encryption (HE).**   Our implementation is based on the C++ HEaaN library (CryptoLab, 2022). All of our experiments used $8 \times$ Nvidia GeForce RTX 4090 24GB GPUs. We use the FGb parameter of the HEaaN library with the specific values in Table 5 (Appendix B). Overall, we can do $L = 9$ (p)Mults before BTS, which means the minimum required level for BTS is 3. The input range of BTS is $[-1, 1]$. If the input range exceeds it, we need another BTS algorithm; ExtBTS is an extended bootstrapping for having a wide input range $[-2^{20}, 2^{20}]$, which require at least 4 levels to operate. We usually use ExtBTS in our HE model except when the input range is guaranteed. For the detailed parameter information and time measurement of HE operations, refer to Table 5 in Appendix B. Note that Mult is $6\times$ slower than pMult and Rot is $24\times$ slower than pRot. These are the main reasons for CCMM is much slower than PCMM.

**Architecture.**   We use an encoder-only transformer in the style of BERT (Devlin et al., 2019) and largely follow the setup of (Geiping & Goldstein, 2023). We remove all bias terms in the linear layers of transformer blocks and use the same tokenization as in (Geiping & Goldstein, 2023). The hidden dimensions of embedding, attention layers, and FFNs are 768. The number of attention heads is 12. In FFNs, we enlarge the hidden dimension into 3072 with a dense layer and reduce that to 768 using a gated linear unit (GLU) (Dauphin et al., 2017) and ReLU. In contrast to (Geiping & Goldstein, 2023), we modify the dense layer of the shape $(768, 1024)$ in the classification head attached to the BERT into the two dense blocks of the shapes $(768, 32)$ and $(32, 1024)$, which allows us to use more optimized CCMMs (see Appendix D.2). We set the number of transformer layers as 2 for the practical computation time. We apply LoRA only to the query, value, and key layers as applying LoRA to other layers (e.g., FFN) did not give a noticeable performance gain in our experiments. LoRA rank is 2 for all LoRA layers.

**Non-polynomial approximation.**   Our transformer implementation under HE requires the polynomial approximation of several non-polynomial functions. We use the following methods for the approximation: Remez approximation (Remez, 1934), Newton's method (Nocedal & Wright, 2006), iterative method, and composition of minimax functions. Remez algorithm is computed using the Sollya tool (Chevillard et al., 2010). For inverse square root functions, we combine Remez approximation with a few Newton's steps. For each non-polynomial approximation method and its precision based on input range and degree, refer to Table 13 in Appendix F. We also classify polynomials used for each downstream task through the HE experiment in the same section.

### 5.2 TIME MEASURMENTS

We now present the speedups provided by our proposed methods. We use an input token length of 128. For further details on the optimizations of homomorphic matrix multiplication and LoRA-friendly CCMM, refer to Appendix D.1 and D.2.

**CCMM vs. PCMM.**   As explained in Appendix C and shown by the time differences between ciphertext and plaintext operations in Table 5 (Appendix E), CCMM is approximately $5\times$ slower than PCMM for the same matrix sizes evaluation (Table 1a).

Table 1: Computation time comparison for homomorphic matrix multiplication and Softmax with GK. For matrix multiplication, we assume the evaluation of matrices for $128 \times 768$ by $768 \times 768$ where $128 \times 768$ is in ciphertext, and $768 \times 768$ is either in ciphertext or plaintext. PCMM and GK are much faster than their respective comparison group.

<div>

(a) CCMM vs. PCMM.

|      | Time (s) | Factor |
|------|----------|--------|
| CCMM | 1.259    | 1      |
| PCMM | 0.275    | **4.58** |

(b) Softmax vs. GK

|         | Time (s) | Factor |
|---------|----------|--------|
| Softmax | 8.99     | 1      |
| GK      | 1.44     | **6.24** |

</div>

**Softmax vs. GK.**   In Table 1b, we compare the computation time of Softmax and GK under HE, where the input range is $[-2^{10}, 0]$ and we evaluate 12 pairs of $128 \times 128$ matrices row-wise for softmax and on matrices of the same size for GK. For the softmax function, we utilize the method of HETAL (Lee et al., 2023b), which approximates the maximum function, the exponential function, and the inverse function homomorphically. GK is much faster than the softmax function evaluation using the HETAL (Lee et al., 2023b) approach under the same experimental setting.

**Optimizer.**   In the experiments, we use $\mathrm{AdamW\text{-}HE}$, a modified version of $\mathrm{AdamW}$ (Loshchilov & Hutter, 2019) is stated in the following as Algorithm 1.

---

**Algorithm 1** $\mathrm{AdamW\text{-}HE}$

---

1: **Initialize:** $m_0 \leftarrow 0, v_0 \leftarrow 0$
2: **procedure** $\mathrm{AdamW\text{-}HE}(\gamma, \beta_1, \beta_2, \varepsilon > 0, \lambda, \theta_{t-1}, m_{t-1}, v_{t-1}, g_t)$
3: $\quad \theta_t \leftarrow \theta_{t-1} - \gamma \lambda \theta_{t-1}$
4: $\quad m_t \leftarrow \beta_1 m_{t-1} + (1 - \beta_1) g_t$
5: $\quad v_t \leftarrow \beta_2 v_{t-1} + (1 - \beta_2) g_t^2$
6: $\quad \widehat{m_t} \leftarrow m_t / (1 - \beta_1^t)$
7: $\quad \widehat{v_t} \leftarrow v_t / (1 - \beta_2^t)$
8: $\quad \theta_t \leftarrow \theta_t - \gamma \widehat{m_t} / \sqrt{\widehat{v_t} + \varepsilon} \qquad \left( \text{In AdamW}, \theta_t \leftarrow \theta_t - \gamma \widehat{m_t} / \left( \sqrt{\widehat{v_t}} + \varepsilon \right) \right)$
9: $\quad$ **Return** $\theta_t$
10: **end procedure**

---

In the original version of AdamW, the Step 8 has the form $\theta_t \leftarrow \theta_t - \gamma \widehat{m_t} / (\sqrt{\widehat{v_t}} + \varepsilon)$, and this is numerically challenging to evaluate for the following two reasons:

- To calculate $1/\left(\sqrt{\widehat{v_t}} + \varepsilon\right)$, we have to approximate $\sqrt{x}$ and $1/x$, which are delicate.

- Conventionally, $\varepsilon > 0$ is set to a sufficiently small value, such as $10^{-12}$. However, approximating $1/x$ on the large range $[\varepsilon, 1]$ is challenging if $\varepsilon$ is small.

Therefore, we change Step 8 in the Algorithm 1 into

$$\theta_t \leftarrow \theta_t - \gamma \widehat{m_t} / \sqrt{\widehat{v_t} + \varepsilon},$$

and we choose a value of $\varepsilon > 0$ that is not too small by considering both the performance and the approximation accuracy. Also, we use the maximum-dividing trick as in (3). These modifications allow $\mathrm{AdamW\text{-}HE}$ to be run stably under HE. Refer to Table 11 in Appendix E for hyperparameters used in $\mathrm{AdamW\text{-}HE}$ for each task.

**Computation time for our model and downstream tasks.**   Table 2 presents the execution times of each setup. We observe a speedup of $6.94\times$ for fine-tuning and $2.3\times$ for inference. Table 3 compares the execution time for each downstream task with the Full+SM model. Our model consistently achieves a $4\times$ improvement across each downstream task. For the detailed end-to-end time measurement for each block operation of our target model, see Table 12 in Appendix E.

Table 2: Comparison of fine-tuning and inference times for full fine-tuning with Softmax (Full+SM), LoRA fine-tuning with Softmax (LoRA+SM), and LoRA fine-tuning with GK (LoRA+GK, Ours) approaches under a single GPU. Our model achieves speedup in both fine-tuning and inference.

| Time (s) | Fine-tuning | | | Inference | | |
| --- | --- | --- | --- | --- | --- | --- |
| | Full+SM | LoRA+SM | LoRA+GK (Ours) | Full+SM | LoRA+SM | LoRA+GK (Ours) |
| 2 layers | 169.2 | 65.16 | 49.22 | 61.12 | 41.72 | 25.78 |
| Class. head | 1.52 | 1.5 | 1.5 | 0.72 | 0.72 | 0.72 |
| Optimizer | 252.83 | 10.31 | 10.31 | - | - | - |
| Total | 423.55 | 76.97 | 61.03 | 61.84 | 42.44 | 26.5 |
| Factor | 1 | 5.5 | **6.94** | 1 | 1.46 | **2.33** |

Table 3: Execution time analysis of fine-tuning for each downstream task, comparing our method with Full+SM. Our model shows the consistent $4\times$ improvement for each downstream task.

| | Task | RTE | MRPC | STS-B | COLA | SST-2 | QNLI |
| --- | --- | --- | --- | --- | --- | --- | --- |
| Time per epoch (h) | Ours | 4.8 | 7.1 | 11.1 | 16.5 | 130 | 202 |
| | Full+SM | 19.5 | 28.8 | 45.1 | 67.1 | 529 | 822 |

## 5.3 DOWNSTREAM TASK PERFORMANCE ON ENCRYPTED LANGUAGE MODELS

We evaluate our model on the GLUE benchmark (Wang et al., 2018). However, we exclude WMNI (Devlin et al., 2019) as in Geiping & Goldstein (2023), and MNLI (Williams et al., 2018) and QQP (Wang et al., 2018) due to their size and our computational constraints. We fine-tune using the cross-entropy loss for tasks including CoLA (Warstadt et al., 2019), MRPC (Dolan & Brockett, 2005), RTE (Giampiccolo et al., 2007), QNLI (Wang et al., 2018), and SST-2 (Socher et al., 2013), and MSE loss for STSB (Cer et al., 2017). We fine-tune SST-2 and QNLI only for 1 epoch and 5 epochs for others since the former are more computationally heavy. One can see the scores for Full+SM, LoRA+GK under plaintext, and LoRA+GK under HE (ours) in Table 4. We set $\varepsilon = 10^{-12}$ in the optimizer for all plaintext settings. For plaintext experiments, we repeat 10 times for each task and choose the best score. For experiments under HE, different $\varepsilon$ values are chosen for each task. The results in Table 4 indicate that using LoRA, Gaussian kernel, and homomorphic encryption results in only a minimal reduction in model accuracy compared to using full fine-tuning, softmax, and no encryption.

Table 4: GLUE results on our homomorphically encrypted language model. "Matthews corr." means Matthews correlation. "Full+GK" denotes full fine-tuning with GK. For all metrics, higher values mean better performance. Our model performs comparably to plaintext models.

| Task | Plaintext Fine-tuning | | | LoRA+GK under HE (Ours) | |
| --- | --- | --- | --- | --- | --- |
| | Full+SM | Full+GK | LoRA+GK | Eval under Plaintext | Eval under HE |
| CoLA (Matthews corr. ↑) | 0.2688 | 0.2640 | 0.1883 | 0.1512 | 0.1575 |
| MRPC (F1 ↑) | 0.8304 | 0.8431 | 0.8258 | 0.8147 | 0.8147 |
| RTE (Accuracy ↑) | 0.5884 | 0.6101 | 0.5776 | 0.5957 | 0.5993 |
| STSB (Pearson ↑) | 0.8164 | 0.8215 | 0.8107 | 0.8002 | 0.7997 |
| SST-2 (Accuracy ↑) | 0.8991 | 0.8911 | 0.8567 | 0.8188 | 0.8188 |
| QNLI (Accuracy ↑) | 0.8375 | 0.8287 | 0.8040 | 0.7827 | 0.7827 |
| Average | 0.7068 | 0.7098 | 0.6772 | 0.6606 | 0.6621 |

## 6 CONCLUSION

In this work, we present a homomorphic encryption (HE) friendly transformer architecture and experimentally demonstrate that an encrypted BERT-style transformer exhibits significant speedups in both fine-tuning and inference under HE. Our architecture specifically resolves two computational bottlenecks using LoRA and Gaussian kernel (GK).

The field of privacy-preserving machine learning using large neural networks will advance through research on (i) training small, efficient language models with considerations unrelated to encryption, (ii) improving the computational efficiency of fundamental computational primitives under HE with considerations unrelated to machine learning, and (iii) developing machine learning models tailored toward the encrypted usage. This work represents the first steps of progress in the direction (iii).

## ACKNOWLEDGEMENTS

This work has been supported by the National Research Foundation of Korea (NRF) under a grant funded by the Korea government (MSIT) (No. 2022R1A5A6000840). Jung Hee Cheon and Taeseong Kim have been supported by Samsung Electronics Co., Ltd (IO201209-07883-01).

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

# A  DIAGRAM: FULL FINE-TUNING VS. LORA FINE-TUNING IN BERT

Figure 2, 3 illustrate the complete diagram of our target model with respect to full fine-tuning and LoRA fine-tuning, respectively. Orange boxes indicate where updates occur during fine-tuning, and $c$ in the classification head layer denotes the number of classes. In full fine-tuning, large-size weight matrices are updated and transformed from plaintexts to ciphertexts, which means the large-size CCMMs must be processed during both fine-tuning and inference. In contrast, only small-size weight matrices are transformed into ciphertexts in LoRA fine-tuning.

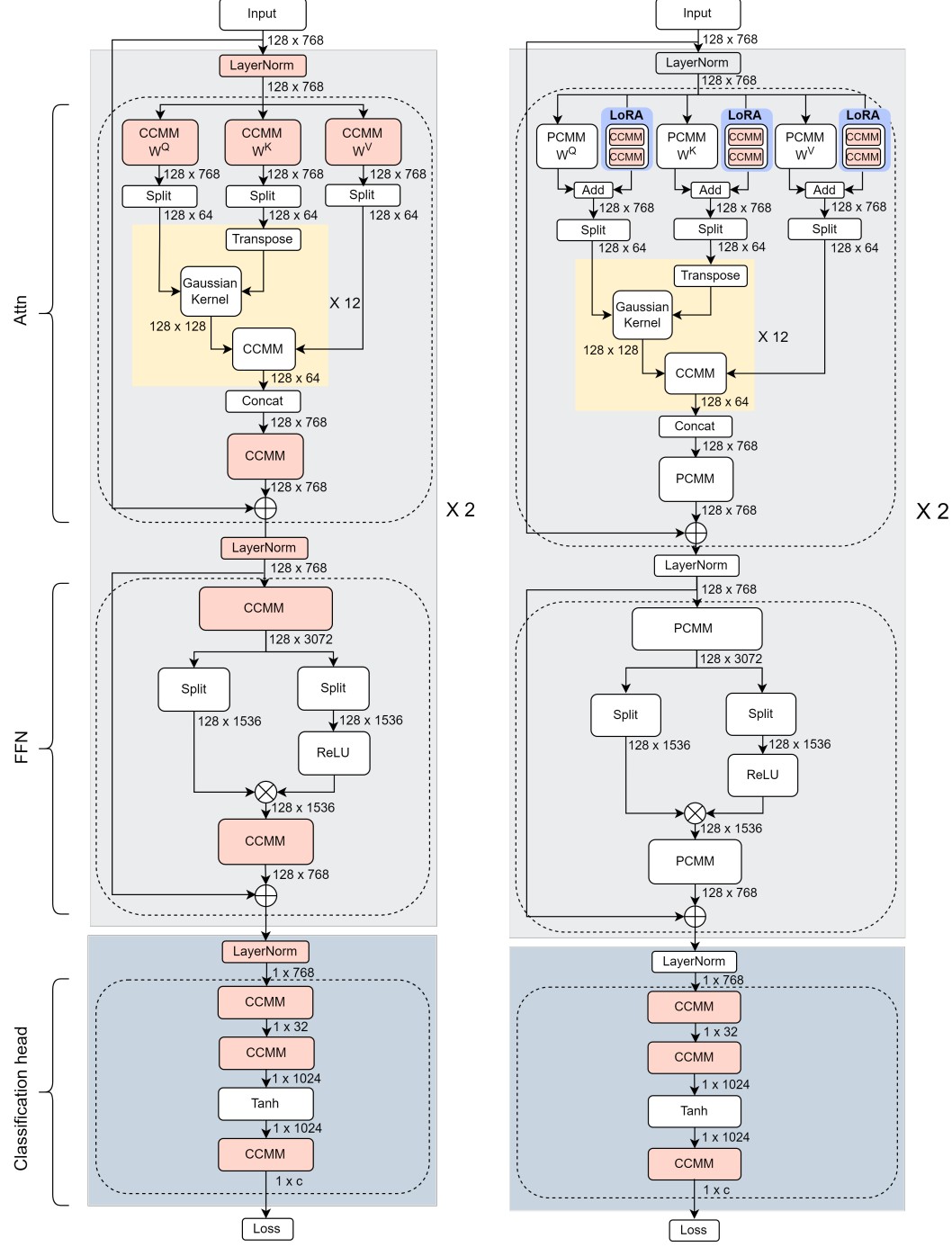

Figure 2: Full fine-tuning.  Figure 3: LoRA fine-tuning.

# B CKKS FUNCTIONALITES

The CKKS (Cheon et al., 2017) is a homomorphic encryption scheme that allows one to encrypt real or complex data as a message polynomial and perform approximate arithmetic on the encrypted message. More precisely, CKKS scheme supports a single-instruction-multiple-data (SIMD) property from encoding (Ecd)/decoding (Dcd) map: given a power-of-two positive integer ring degree $N$, encoding map is defined as $\mathsf{Ecd} : \mathbb{C}^{N/2} \to \mathcal{R}_Q = \mathbb{Z}_Q[x]/(x^N + 1)$, where $Q$ is a positive integer. The decoding map Dcd can be interpreted as an inverse process of Ecd. Ecd/Dcd represents a conversion between $(\mathbb{C}^{N/2}, \oplus, \odot)$ and $(\mathcal{R}_Q, +, \cdot)$, where $\oplus$ and $\odot$ are component-wise addition and multiplication, respectively. In the CKKS scheme, each of Ecd and Dcd is processed before encryption or after decryption, respectively. Encryption ($\mathsf{Enc_{pk}}$) and Decryption ($\mathsf{Dec_{sk}}$) are a ring learning with errors (RLWE) (Stehlé et al., 2009; Lyubashevsky et al., 2010) encryption and decryption, respectively. We can operate addition/multiplication on those vectors via polynomial operations from the ring isomorphism. Here, we use $\approx$ notation to indicate approximate equality, as the CKKS scheme evaluates approximately (for more details, refer to Cheon et al. (2017));

- **Key generation**: Given a security parameter $\lambda$ and a subset $S \subset \{1, 2, \cdots, N/2\}$, return a public key pk, a secret key sk, a relinearization key rlk, and rotation keys $\{\mathsf{rk}_i\}_{i \in S}$.
- **Encryption**: Given pk and a message $\mathbf{m}$, return a ciphertext $\mathrm{ct} = \mathsf{Enc_{pk}}(\mathbf{m}) = \mathsf{Enc}(\mathsf{Ecd}(\mathbf{m}), \mathsf{pk})$.
- **Decryption**: Given sk and a ciphertext ct, return a message $\mathbf{m}' = \mathsf{Dec_{sk}}(\mathrm{ct}) = \mathsf{Dcd}(\mathsf{Dec}(\mathrm{ct}, \mathsf{sk}))$, where $\mathbf{m}' \approx \mathbf{m}$.
- **Addition**: Given two ciphertexts $\mathrm{ct}_1$ and $\mathrm{ct}_2$, return $\mathrm{ct}_{add} = \mathsf{Add}(\mathrm{ct}_1, \mathrm{ct}_2)$, satisfying $\mathsf{Dec_{sk}}(\mathrm{ct}_{add}) \approx \mathsf{Dec_{sk}}(\mathrm{ct}_1) \oplus \mathsf{Dec_{sk}}(\mathrm{ct}_2)$. Addition also can be evaluated between plaintext and ciphertext.
- **Multiplication**: Given two ciphertexts $\mathrm{ct}_1$ and $\mathrm{ct}_2$ and a linearization key rlk, return $\mathrm{ct}_{mult} = \mathsf{Mult}(\mathrm{ct}_1, \mathrm{ct}_2)$, satisfying $\mathsf{Dec_{sk}}(\mathrm{ct}_{mult}) \approx \mathsf{Dec_{sk}}(\mathrm{ct}_1) \odot \mathsf{Dec_{sk}}(\mathrm{ct}_2)$.
- **Multiplication by Plaintext**: Given a plaintext pt and a ciphertext ct, return $\mathrm{ct}_{pmult} = \mathsf{pMult}(\mathrm{pt}, \mathrm{ct})$, satisfying $\mathsf{Dec_{sk}}(\mathrm{ct}_{pmult}) \approx \mathsf{Dcd}(\mathrm{pt}) \odot \mathsf{Dec_{sk}}(\mathrm{ct}_{pmult})$.
- **Rotation**: Given a rotation key $\mathsf{rk}_i$ for $i \in S$ and a ciphertext ct with $\mathsf{Dec_{sk}}(\mathrm{ct}) \approx (m_0, m_1, \ldots, m_{N/2-1})$, return a ciphertext $\mathrm{ct}_{rot_i} = \mathsf{Rot}(\mathrm{ct}, i)$ satisfying $\mathsf{Dec_{sk}}(\mathrm{ct}_{rot_i}) \approx (m_i, \ldots, m_{N/2-1}, m_0, \ldots, m_{i-1})$. The rotation index $i$ acts modulo $N/2$, so $\mathsf{Rot}(\cdot, -i) = \mathsf{Rot}(\cdot, N/2 - i)$. pRot is the same operation on the plaintext side.

**HE parameter and its time measurement.** During HE implementation, we use the FGb parameter, one of the HEaaN-providing parameters. For the detailed value of the FGb parameter, refer to Table 5. We also list the time measurement of HE operations under a single GPU. We know that Mult and Rot, operated in ciphertext, are slower than pMult and pRot, which are related to plaintext operations, respectively.

Table 5: The FGb parameter and time measurement for each HE operation under the single GPU. The terms $\log(QP), N, L, \lambda$ denote the bit lengths of the largest modulus, ring degree, multiplicative depth, and security parameter.

| FGb parameter | | | | | Time measurement of each homomorphic operation (ms) | | | | | | |
|---|---|---|---|---|---|---|---|---|---|---|---|
| $\log(QP)$ | $N$ | $L$ | $h$ | $\lambda$ | Add | pRot | Rot | BTS | ExtBTS | pMult | Mult |
| 1555 | $2^{16}$ | 9 | 192 | 128 | 0.02 | 0.02 | 0.49 | 60 | 137 | 0.1 | 0.6 |

# C HOMOMORPHIC MATRIX MULTIPLICATION ALGORITHM

Consider $d \times d$ homomorphic matrix multiplication. JKLS (Jiang et al., 2018) suggests the fastest known HE square matrix multiplication algorithm with $\mathcal{O}(d)$ HE computational complexity (for more precise number of each operation, refer to Table 6) when the matrix elements can be packed in a ciphertext, specifically when $d^2 \leq N/2$. The required HE operations include Add, (p)Rot, and (p)Mult. The computation time of Rot and Mult operations differ between plaintext and ciphertext (see Table

5), with operations in ciphertext being significantly slower. Consequently, CCMM is much slower than PCMM; the number of different message space operations is exactly $d + 2\sqrt{d}$ for one homomorphic matrix multiplication. For precise time difference between two kinds of homomorphic matrix multiplication, see Section 5.2.

When $d^2 > N/2$, where we cannot encrypt the matrix into one ciphertext, we can also utilize the JKLS algorithm by computing block-wise matrix multiplication. For example, if we take $N = 2^{16}$ for the base ring degree, we can pack two $128 \times 128$ matrices into one ciphertext. When $d = 768$, we require 18 ciphertexts to pack $d \times d$ matrix. After packing each block matrices into ciphertexts, we repeat the corresponding block-wise matrix multiplication to all the weights. Thus, for large-size matrix multiplications, the computation time gap between CCMM and PCMM will be proportional to the number of ciphertexts required for packing.

In addition, JKLS introduced a homomorphic rectangular matrix multiplcation of size $\ell \times d$ by $d \times d$ or vice-versa, with $\ell < d$ and $\ell d \leq N/2$. We can use this method to evaluate LoRA CCMMs, where we need to evaluate the homomorphic rectangular matrix multiplication sequentially. For more detailed HE operations numbers, see Table 7.

Table 6: The number of each HE operation for one $d \times d$ homomorphic matrix multiplication required by JKLS (Jiang et al., 2018) algorithm.

| Operations | Add | pMult | Rot | Mult | Depth |
|---|---|---|---|---|---|
| JKLS (Jiang et al., 2018) | $6d$ | $4d$ | $3d + 5\sqrt{d}$ | $d$ | 3 |

Table 7: The number of each HE operation for rectangular homomorphic matrix multiplication $\ell \times d$ by $d \times d$ required by JKLS (Jiang et al., 2018) rectangular matrix algorithm.

| Operations | Add | pMult | Rot | Mult | Depth |
|---|---|---|---|---|---|
| JKLS (Jiang et al., 2018) | $3d + 2\ell + \log(d/\ell)$ | $3d + 2\ell$ | $3\ell + 5\sqrt{d} + \log(d/\ell)$ | $\ell$ | 3 |

## D  OPTIMIZATIONS FOR HOMOMORPHIC MATRIX MULTIPLICATION

The CKKS scheme packs all input data into a polynomial from encoding (Ecd). The packed data are assigned to fixed positions within the polynomial, and to change these positions, we have to use the (p)Rot operation. In the homomorphic matrix multiplication algorithm, several (p)Rot operations are required. Our target model has sequential matrix multiplications, so it is more efficient to fix the packing structure. Among various packing methods, we use row-wise packing (see Figure 4a) for all packed data except for LoRA weights, which will be explained in the next subsection. Because using the FGb parameter can pack a total $2^{15}$ data in one message space, we consider one plain/ciphertext having $2^{15}$ data, which corresponds to the size either $128 \times 256$ or $256 \times 128$ in our implementation. Thus, during our model computation, we will assume one plain/ciphertext is packed $128 \times 256$ matrix row-wise.

### D.1  HOMOMORPHIC MATRIX MULTIPLICATION

Here, we introduce the optimization techniques used in our homomorphic matrix multiplication implementation. We adopt three optimization techniques to improve the speed of homomorphic matrix multiplication:

- **Evalation in lower level:**  Because homomorphic matrix multiplication contains numerous HE operations and its speed depends on the current level, the ciphertexts containing matrix values are brought down to the minimum required level before performing the evaluation (Bae et al., 2024; Park et al., 2024), even if BTS may be needed for the next block evaluation. We adhere to this strategy in our homomorphic matrix multiplication by taking the JLKS (Jiang et al., 2018) algorithm, which requires 3 levels. Consequently, evaluating the bottom-level (7 level) PCMM is

2.35× faster than the top-level (12 level) PCMM (refer to Table 8) of size $128 \times 256$ by $256 \times 128$. This difference is even more pronounced in CCMM.

- **Pre-computation and reuse for left-matrix:** When we perform homomorphic matrix multiplication of size $128 \times 768$ by $768 \times 768$, we need to split the matrices as three and eighteen block matrices (see Figure 4c), respectively. And we go through each block matrix multiplication to get a corresponding result. From the property of block matrix multiplication, we know the same left matrices are used several times to multiply by the right block matrices. Thus, we do pre-computation using JKLS (Jiang et al., 2018) and save these objects for left-matrices. Using this strategy, we can reduce the total number of JKLS algorithm calls, which require numerous HE operations (as shown in Table 6), from 36 to 21. This is a trade-off between memory and computation time.

- **Lazy key-switching/rescaling:** When a ciphertext is evaluated by an automorphism, we need to do a key-switching operation. This is an expensive process. If the result ciphertext is made from several ciphertexts resulting in the same automorphism, we are used to take *lazy* key-switching (hoisting) (Halevi & Shoup, 2018); that is, we go through only one key-switching after collecting all results instead of doing key-switching for each automorphism. By taking lazy key-switching, we can speed up the required computation time. Lazy rescaling is also a similar strategy for reducing computation time and added errors when doing a rescaling operation, which is an operation contained in Mult.

Table 8: Computation time comparison between PCMMs according to the different levels. The lower-level homomorphic matrix multiplcation is faster than the top-level evaluation.

| PCMM | Time (s) | Factor |
|---|---|---|
| 12 level | 0.242 | 1 |
| 7 level | 0.103 | **2.35** |

(a) Row-wise packing  (b) Zero-padding packing

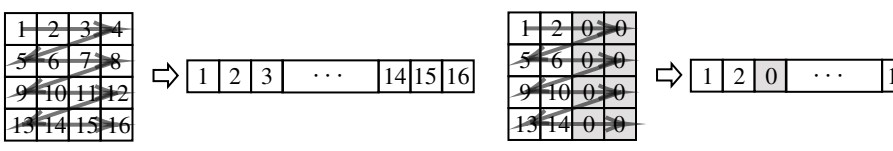

(c) Block-wise matrix multiplication

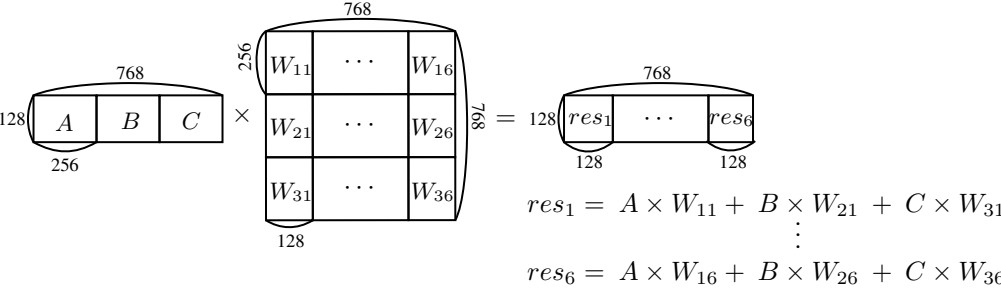

$$res_1 = A \times W_{11} + B \times W_{21} + C \times W_{31}$$
$$\vdots$$
$$res_6 = A \times W_{16} + B \times W_{26} + C \times W_{36}$$

Figure 4: Row-wise packing method for matrix representation, utilizing zero-padding for non-square matrices, followed by block-wise matrix multiplication for efficient processing of large matrices.

## D.2 LORA STYLE CCMM

LoRA fine-tuning consists of two CCMMs. Because the LoRA weights are rectangular matrices, we need to do zero-padding (see Figure 4b) for the second weight to make it a square matrix. After that, we can follow the optimized homomorphic rectangular matrix multiplication algorithm sequentially, which we call the *Rectangular MM* algorithm in the following. However, when we consider that our

target LoRA rank(=2) is quite small, this Rectangular MM approach generates an inefficiency since there are many non-usable spaces in the ciphertext from zero-padding. Thus, we make a more efficient algorithm for LoRA CCMMs than following the Rectangular MM. Our optimized algorithm can be applied in cases similar to LoRA CCMMs, which involve sequential matrix multiplication where one dimension is significantly smaller than the other. In our model, we also use these optimized CCMMs during classification head layer evaluation, where we have to perform homomorphic rectangular matrix multiplications of size $1 \times 768$ by $768 \times 32$ and $1 \times 32$ by $32 \times 1024$. For convenience, this subsection will describe the case regarding LoRA CCMMs with rank 2: $128 \times 768$ by $768 \times 2$ and $128 \times 2$ by $2 \times 768$, sequentially.

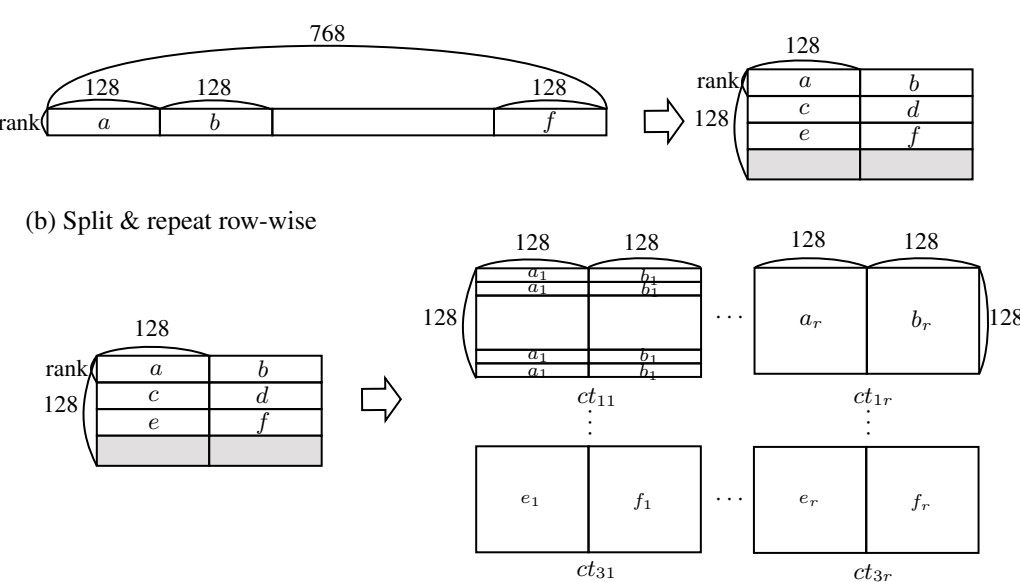

Figure 5: LoRA-friendly packing is used when the given matrix has one long and one short size. Split & repeat row-wise divides and copies each row into ciphertexts, which is used during LoRA CCMMs and $a_i$ denotes $i$-th row of matrix $a$. The shaded block matrices represent zero-padded blocks.

### D.2.1 LoRA-FRIENDLY PACKING

In our target model, the LoRA weights are either $768 \times$ rank or rank $\times 768$. Both weights have one long and one short dimension. Instead of directly following row-wise packing used in entire message packing, we pack LoRA weights with another approach. Figure 5a represents the packing method, called LoRA-friendly packing, used in LoRA weights. Even if $768 \times$ rank is a column rectangular matrix shape, we follow the LoRA-friendly packing by considering the transposed column rectangular matrix. This does not mean we evaluate homomorphic transposition, where we use an algorithm in JKLS (Jiang et al., 2018) that also has numerous HE operations. But treat its column rectangular matrix with the corresponding order in a row-wise packed structure:

$$\{A_{ij}\}_{1 \leq i \leq 768, 1 \leq j \leq \text{rank}} \mapsto \{B_k\}_{1 \leq k \leq N/2} = \begin{cases} A_{ij} & \text{if } k = i + 768 * (j-1) \\ 0 & \text{otherwise} \end{cases}$$

By following this, we can remove the required evaluations of transposition during backpropagation computation.

### D.2.2 OPTIMIZED HOMOMORPHIC LoRA MATRIX MULTIPLICATION ALGORITHM

We optimize the sequential LoRA CCMMs in Algorithm 5. This algorithm consists of several HE operations and three operation blocks: (i) Split & repeat row-wise (Algorithm 2) for weights, (ii) repeat column-wise (Algorithm 3), and (iii) collect into the first column (Algorithm 4).

---

**Algorithm 2** Split & repeat row-wise (Figure 5b)

---

1: **Input:** A ciphertext $\text{ct}_w$ having LoRA weight, LoRA rank $r$, plaintext $M_{row}$ encoding $128 \times 256$ matrix where all entries are zero except for the first row, which is entirely ones.
2: **Output:** Ciphertexts $\{\text{ct}_{ij}\}_{i \in \{1,2,3\}, j \in \{1, \cdots, r\}}$.
3: **procedure** SplitRepeat($\text{ct}_w, r, M_{row}$)
4:     **for** $i \in \{1, 2, 3\}$ **do**
5:         $\text{tmp} \leftarrow \text{ct}_w$
6:         **if** $i$ is not 1 **then**
7:             $\text{tmp} \leftarrow \text{Rot}(\text{tmp}, (i-1) \times r \times 256)$
8:         **end if**
9:         **for** $j \in \{1, \cdots, r\}$ **do**
10:             **if** $j$ is not 1 **then**
11:                 $\text{tmp} \leftarrow \text{Rot}(\text{tmp}, (j-1) \times 256)$
12:             **end if**
13:             $\text{ct}_{ij} \leftarrow \text{pMult}(M_{row}, \text{tmp})$
14:             **for** $k \in \{0, \cdots, \log_2(128) - 1\}$ **do**
15:                 $\text{tmp} \leftarrow \text{Rot}(\text{ct}_{ij}, -2^k \times 256)$
16:                 $\text{ct}_{ij} \leftarrow \text{Add}(\text{ct}_{ij}, \text{tmp})$
17:             **end for**
18:         **end for**
19:     **end for**
20:     **Return** $\{\text{ct}_{ij}\}_{i \in \{1,2,3\}, j \in \{1, \cdots, r\}}$
21: **end procedure**

---

**Algorithm 3** Repeat column-wise

---

1: **Input:** Ciphertexts $\{\text{ct}_i\}_{i \in \{1, \cdots, r\}}$, LoRA rank $r$.
2: **Output:** Ciphertexts $\{\text{ct}_i\}_{i \in \{1, \cdots, r\}}$.
3: **procedure** RepeatCol($\{\text{ct}_i\}_{i \in \{1, \cdots, r\}}, r$)
4:     **for** $i \in \{1, \cdots, r\}$ **do**
5:         **for** $k \in \{0, \cdots, \log_2(256) - 1\}$ **do**
6:             $\text{tmp} \leftarrow \text{Rot}(\text{ct}_i, -2^k)$
7:             $\text{ct}_i \leftarrow \text{Add}(\text{ct}_i, \text{tmp})$
8:         **end for**
9:     **end for**
10:     **Return** $\{\text{ct}_i\}_{i \in \{1, \cdots, r\}}$
11: **end procedure**

---

**Algorithm 4** Collect into the first column

---

1: **Input:** A ciphertext $\{a_{ij}\}_{i \in \{1,2,3\}, j \in \{1, \cdots, r\}}$, LoRA rank $r$, plaintext $M_{col}$, encoding $128 \times 256$ matrix where all entries are zero except for the first column, which is entirely ones.
2: **Output:** Ciphertexts $\{\text{ct}_j\}_{j \in \{1, \cdots, r\}}$.
3: **procedure** CollectFirstCol($\text{ct}_{ij}, r, M_{col}$)
4:     **for** $j \in \{1, \cdots, r\}$ **do**
5:         $\text{ct}_j \leftarrow a_{1j}$
6:         **for** $i \in \{2, 3\}$ **do**
7:             $\text{ct}_j \leftarrow \text{Add}(\text{ct}_j, a_{ij})$
8:         **end for**
9:         **for** $k \in \{0, \cdots, \log_2(256) - 1\}$ **do**
10:             $\text{tmp} \leftarrow \text{Rot}(\text{ct}_j, 2^k)$
11:             $\text{ct}_j \leftarrow \text{Add}(\text{ct}_j, \text{tmp})$
12:         **end for**
13:         $\text{ct}_j \leftarrow \text{pMult}(M_{col}, \text{ct}_j)$
14:     **end for**
15:     **Return** $\{\text{ct}_j\}_{j \in \{1, \cdots, r\}}$
16: **end procedure**

---

---

**Algorithm 5** LoRA CCMMs

---

1: **Input:** Ciphertexts $\{ct_i\}_{i \in \{1,2,3\}}$, LoRA weight ciphertexts $ct_A$ and $ct_B$, LoRA rank $r$, plaintexts $M_{row}, M_{col}$.
2: **Output:** Result ciphertexts $\{res_i\}_{i \in \{1,2,3\}}$ of LoRA CCMMs.
3: **procedure** LoRA CCMMs($\{ct_i\}_{i \in \{1,2,3\}}, ct_A, ct_B, r, M_{row}, M_{col}$)
4:     $\{A_{ij}\} \leftarrow$ SplitRepeat($ct_A, r, M_{row}$)
5:     **for** $i \in \{1, 2, 3\}$ **do**
6:         **for** $j \in \{1, \cdots, r\}$ **do**
7:             $W_{ij} \leftarrow$ Mult($A_{ij}, ct_i$)
8:         **end for**
9:     **end for**
10:     $\{tmp_k\} \leftarrow$ CollectFirstCol($\{W_{ij}\}, r, M_{col}$)
11:     $\{tmp_k\} \leftarrow$ RepeatCol($\{tmp_k\}, r$)
12:     $\{B_{ij}\} \leftarrow$ SplitRepeat($ct_B, r, M_{row}$)
13:     **for** $i \in \{1, 2, 3\}$ **do**
14:         $res_i \leftarrow$ Mult($B_{i1}, tmp_1$)
15:         **for** $j \in \{2, \cdots, r\}$ **do**
16:             $B_{ij} \leftarrow$ Mult($B_{ij}, tmp_j$)
17:             $res_i \leftarrow$ Add($B_{ij}, res_i$)
18:         **end for**
19:     **end for**
20:     **Return** $\{res_i\}_{i \in \{1,2,3\}}$
21: **end procedure**

---

### D.2.3 COMPARISON RECTANGULAR MM AND OPTIMIZED LORA CCMMS

We compare two approaches, Rectangular MM and our optimized LoRA CCMMs (Algorithm 5). We list up the required number of each HE operation for each method (Table 9) based on the given number of JKLS (Jiang et al., 2018) algorithm (Table 7) and the computation time comparison in implementation (Table 10). Our optimized LoRA CCMMs are $4.45\times$ faster than Rectangular MM. In addition, using LoRA-friendly packing, we can reduce the required number of homomorphic transposition evaluations.

Table 9: The total required number of HE operations for $128 \times d$ by $d \times r$ and $128 \times r$ by $r \times d$ homomorphic matrix multiplications where $r$ denotes the LoRA rank. Rectangular MM denote a method that takes homomorphic rectangular matrix multiplication sequentially. Algorithm 5 requires much smaller HE operations than following Rectangular MM algorithms.

| Operations | Add | pMult | Rot | Mult | Depth |
|---|---|---|---|---|---|
| Rectangular MM | $18d + 12r + 6\log(d/r)$ | $18d + 12r$ | $18r + 30\sqrt{d} + 6\log(d/r)$ | $6r$ | 6 |
| Algorithm 5 | $42r - 3$ | $4r$ | $40r - 1$ | $6r$ | 3 |

Table 10: Comparison of execution times between Rectangular MM and Algorithm 5. In the figure, MM denotes implementing our homomorphic matrix multiplication, and Tr denotes homomorphic transpose. Algorithm 5 is about $4.45\times$ faster than Rectangular MM.

| | Forward eval. | | Backprop. | | | | |
|---|---|---|---|---|---|---|---|
| | MM | BTS | MM | BTS | Tr | Total | Factor |
| Rectangular MM (s) | 0.65 | 1.3 | 2.84 | 0.42 | 0.71 | 5.92 | 1 |
| Algorithm 5 (s) | 0.12 | 0.76 | 0.29 | 0 | 0.16 | 1.33 | **4.45** |

## E EXPERIMENTAL DETAILS UNDER HE

In this section, we provide experimental details used in our experiments under HE.

**Penalty training** In the experiments, we approximate non-polynomials such as inverse square root in LayerNorm (Ba et al., 2016). However, we cannot assure that the inputs of each non-polynomial lie in the approximation domain. To address this, we charge a penalty as in Baruch et al. (2023) in the pre-training stage. For example, when we approximate ReLU in the $i$th transformer layer on $[-50, 50]$, for an input $X$ of our model, let the input of this ReLU be $X_{i,\text{ReLU}}$. Then we add the term $\mathbf{1}_{\{\|X_{i,\text{ReLU}}\|_\infty > 50\}}$ to the pre-training loss $L_{\text{pre}}$ where $\lambda > 0$. If we let

$$\mathcal{F} = \big\{(f, L_f) \,\big|\, \text{The approximation range of } f \text{ is } [-L_f, L_f]\big\}$$

be the set of all non-polynomials and their approximation domains in our model, then the original loss function is modified into

$$L_{\text{penalty}}(X; \theta) = L_{\text{pre}}(X; \theta) + \lambda \sum_{(f, L_f) \in \mathcal{F}} \mathbf{1}_{\{\|X_f\|_\infty > L_f\}}(X)$$

where $X$ is the input of the model, $\theta$ is the parameters of the model, and $X_f$ is the input of $f \in \mathcal{F}$ for the initial input $X$. We change the original $L_{\text{pre}}$ to $L_{\text{penalty}}$ in the last $k$ (100 in our setting) steps of the pre-training. In this way, we can narrow the input range of non-polynomial functions and are able to approximate them.

**Hyperparameters.** We summarize the hyperparameters used in HE experiments. We first experimented on the plaintext and chose the best hyperparameters for HE experiments.

Table 11: Hyperparameters used for HE experiments. Epsilon means $\varepsilon$ of AdamW-HE in section 5.1. Warmup steps, Number of cycles are used in transformers (Wolf et al., 2020) cosine scheduler, and betas are used in AdamW-HE.

|        | Learning Rate | Epsilon   | Warmup Steps | Number of Cycles | Betas         |
|--------|---------------|-----------|--------------|------------------|---------------|
| CoLA   | $4.0e-3$      | $8.0e-3$  |              |                  |               |
| MRPC   | $5.0e-4$      | $6.0e-4$  |              |                  |               |
| RTE    | $2.0e-4$      | $2.0e-4$  | 0            | 0.5              | $[0.9, 0.999]$ |
| STSB   | $2.5e-2$      | $4.0e-1$  |              |                  |               |
| SST-2  | $8.5e-3$      | $8.0e-4$  |              |                  |               |
| QNLI   | $6.5e-3$      | $8.0e-4$  |              |                  |               |

**End-to-end runtime of 1 layer forward evaluation and backpropagation.** Here, we list all the required block operations runtime in 1 layer case with respect to forward evaluation and backpropagation of LoAR fine-tuning in Table 12. In forward evaluation, the softmax function evaluation is the most time-consuming part in LoRA with Softmax, on the other hand, BTS is the part in LoRA with GK. In both cases of backpropagation, the matrix multiplication time is the most time-consuming part. Thus, replacing the softmax function efficiently and reducing the required number of CCMM are key factors, so we adopt LoRA fine-tuning and the Gaussian Kernel.

Table 12: Lists the end-to-end runtime for each individual operation using single GPU. Tr denotes transposition, and Save is the saving time for backpropagation or optimizer. Softmax evaluation occupies $43\%$, which is the most time-consuming in forward evaluation; however, GK is just $8\%$.

| Forward eval. | Time (s) | Ratio (%) |
|---|---|---|
| PCMM | 1.35 | 6.47 |
| CCMM | 1.87 | 8.96 |
| LayerNorm | 0.48 | 2.3 |
| BTS | 5.74 | 27.53 |
| Softmax | **8.99** | **43.1** |
| ReLU | 1.35 | 6.47 |
| LoRA | 0.85 | 4.07 |
| Tr & Save | 0.23 | 1.1 |
| Total | 20.86 | 100 |

| Backprop. | Time (s) | Ratio (%) |
|---|---|---|
| PCMM | 2.29 | 19.54 |
| CCMM | 3.79 | 32.34 |
| LayerNorm | 0.32 | 2.73 |
| BTS | 4.45 | 37.97 |
| Softmax | **0.02** | **0.17** |
| ReLU | 0.02 | 0.17 |
| LoRA | 0.26 | 2.22 |
| Tr & Save | 0.57 | 4.86 |
| total | 11.72 | 100 |

| Forward eval. | Time (s) | Ratio (%) |
|---|---|---|
| PCMM | 1.35 | 10.47 |
| CCMM | 1.87 | 14.51 |
| LayerNorm | 0.48 | 3.72 |
| BTS | 5.74 | 44.53 |
| GK | **1.02** | **7.91** |
| ReLU | 1.35 | 10.47 |
| LoRA | 0.85 | 6.59 |
| Tr & Save | 0.23 | 1.78 |
| Total | 12.89 | 100 |

| Backprop. | Time (s) | Ratio (%) |
|---|---|---|
| PCMM | 2.29 | 19.54 |
| CCMM | 3.79 | 32.34 |
| LayerNorm | 0.32 | 2.73 |
| BTS | 4.45 | 37.97 |
| GK | **0.02** | **0.17** |
| ReLU | 0.02 | 0.17 |
| LoRA | 0.26 | 2.22 |
| Tr & Save | 0.57 | 4.86 |
| total | 11.72 | 100 |

**Multi-GPU implementation that simulates a batch scenario.** For the LoRA fine-tuning for the downstream tasks, we use 8 GPUs. We follow the same plain model parameters (refer to Appendix E), where the batch size is 16. Because packing multiple datasets to mimic the batch scenario itself has some restrictions (such as large-size weights, long token lengths, and restricted message spaces), we get to use multiple GPUs to make our algorithm keep following batch scenarios. Here, we assign one datum to one GPU. When the weights are updated, the communications between GPUs are raised, and an average of each gradient can be obtained. This speeds up the runtime per epoch, even with a slight communication delay, which is negligible compared to the overall runtime.

# F  UTILIZED POLYNOMIALS FOR EACH DOWNSTREAM TASK

## F.1  OVERVIEW OF POLYNOMIAL APPROXIMATIONS USED IN OUR TARGET MODEL.

Table 13 shows the overview of polynomial approximation utilized in our target model experiments. According to the input range, the optimized approximation method is different. For the inverse square root, we additionally employ a few iterations of Newton's method for the following reason. While minimax approximation optimizes multiplicative depth in polynomial evaluation, it requires numerous multiplications. To mitigate this, we integrate a few Newton iterations, which provide a faster approximation of the inverse square root, though at the cost of additional multiplicative depth.

Table 13: Overview of polynomial approximation methods for various non-polynomial functions with specific usages. "Worst prec." refers to the maximum value of $\|f(x) - poly(x)\|_\infty$ across the input range, while "Avg prec." denotes the average (mean) value. Although we replace the softmax with the Gaussian kernel in attention layers, the softmax function is still used once in computing the cross-entropy loss. Non-polynomials require different polynomial approximations according to the input range.

| | $1/\sqrt{x}$ | $1/x$ | | tanh | | exp | | | ReLU |
|---|---|---|---|---|---|---|---|---|---|
| Input range | [0.0005, 1] | [0.04,1] | [0.8,3.0] | [-5,5] | [-16,16] | [-2,2] | [-13,1] | $[-2^{14},0]$ | [-1,1] |
| Degree | 127 | 63 | 15 | 63 | 127 | 15 | 15 | - | 15/15/27 |
| Worst prec. (bits) | 11.1 | 19.8 | 23.7 | 24.7 | 18.6 | 21.2 | 21.2 | 15 | 10 |
| Avg prec. (bits) | 24.6 | 24.6 | 26.5 | 26.6 | 19.6 | 25.8 | 22.9 | 17.4 | 16.4 |
| Method | Remez + Newton | Remez | | Remez | | Remez | | Iter. | Minimax composition (Lee et al., 2023a) |
| Usage | LayerNorm, AdamW-HE | Loss (Softmax) | | Classification head | | Loss (Softmax) | | GK | FFN |

## F.2 Polynomials for each downstream task

In the previous subsection F.1, we give our polynomial approximation of each non-polynomial. Since the input ranges for non-polynomial functions vary across downstream tasks, appropriate polynomial approximations are required for each task. Below, we list the polynomials used for each task, with Table 14 categorizing them based on their input domains.

- **Inverse square root with additional three Netwon steps**: sqrtInv ([0.0005, 1])
- **Inverse**: inv1 ([0.04, 1]), inv2 ([0.8, 3.0])
- **Exponential**: exp1 ([-2, 2]), exp2 ([-13, 1])
- **Exponential with iterative algorithm**: expIter ($[-2^{14}$,0]) ($p_{14}$ in section 4)
- **ReLU**: relu ([-50, 50])
- **Tanh**: tanh1 ([-5, 5]), tanh2 ([-16, 16])

Table 14: Types of polynomials used in the block operations for each task.

| | LayerNorm | Attn | FFN | Class. head | Loss (Softmax) | AdamW-HE |
|---|---|---|---|---|---|---|
| CoLA MRPC RTE STSB | sqrtInv | expIter | relu | tanh1 | exp1, inv1 | sqrtInv |
| SST-2 QNLI | | | | tanh2 | exp2, inv2 | |

We plot some core polynomial functions used in our work: **expIter**, **sqrtInv**, and **relu**. See Figure 6 and 7. For ReLU and $1/\sqrt{x}$, we slightly modify to use approximations of these functions in the fixed input range. When we approximate $1/\sqrt{x}$, if the input range lies outside the interval [0.0005, 1], we divide a predetermined number $M$ for the inputs to belong in [0.0005, 1]; for an input $x$, we calculate as

$$\frac{1}{\sqrt{x}} = \frac{1}{\sqrt{x/M}} \times \frac{1}{\sqrt{M}}. \tag{3}$$

For ReLU, we set $K > 0$ and for an input $x$ calculate as

$$\text{ReLU}(x) = K\text{ReLU}\left(\frac{x}{K}\right).$$

using the homogeneity of ReLU. In the experiments, we set $K = 50$. You can see in Figure 7 the comparison of these functions and their approximations.

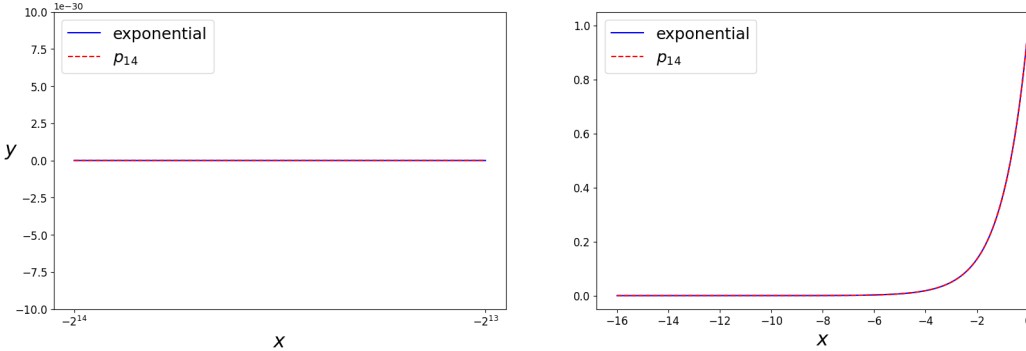

Figure 6: The graph of approximation $p_{14}(= \text{expIter})$ on $\left[-2^{14},\, 0\right]$ of the exponential.

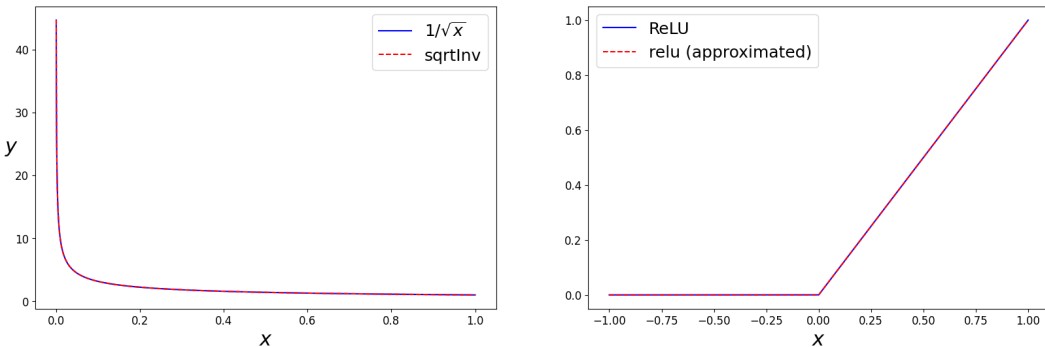

Figure 7: The graphs of approximated inverse square root and $\mathrm{ReLU}$ on $[0.0005,\, 1]$ and $[-1,\, 1]$, respectively.

## G  COMPUTATION ADVANTAGE OF LORA AND GAUSSIAN KERNEL FOR LARGER MODELS

In section 5.2, we saw that CCMM is approximately $4.58 \times$ slower than PCMM when performing matrix multiplication with dimensions $128\times768$ and $768\times768$, where 768 is the embedding size. Also, calculating attention scores using softmax is approximately $6.24 \times$ slower than using GK. However, when the embedding dimension and sequence length become larger, as in GPT-4 (OpenAI, 2023), the gain in computational time by LoRA and GK is amplified. In this section, we calculate the time required to perform PCMM and CCMM and to obtain attention score through softmax and GK as the embedding dimension and the sequence length increase. See the Table 15 for the results.

Table 15: Computation times for performing PCMM and CCMM with $128 \times n$ by $n \times n$ matrices, and evaluating Softmax and GK on a $n$-dimensional input. Factor in each table represents the ratio of computation times between two operations. As the dimension $n$ increases, the performance gains become larger.

(a) Comp. time for PCMM vs. CCMM.

| Dim. ($n$) | 256 | 512 | 768 |
|---|---|---|---|
| PCMM (s) | 0.138 | 0.139 | 0.275 |
| CCMM (s) | 0.297 | 0.463 | 1.259 |
| Factor | 2.15 | 3.33 | 4.58 |

(b) Comp. time for GK vs. Softmax.

| Dim. ($n$) | 128 | 256 | 512 |
|---|---|---|---|
| GK (s) | 0.17 | 0.34 | 1.36 |
| Softmax (s) | 1.3 | 2.86 | 13.04 |
| Factor | 7.65 | 8.41 | 9.59 |

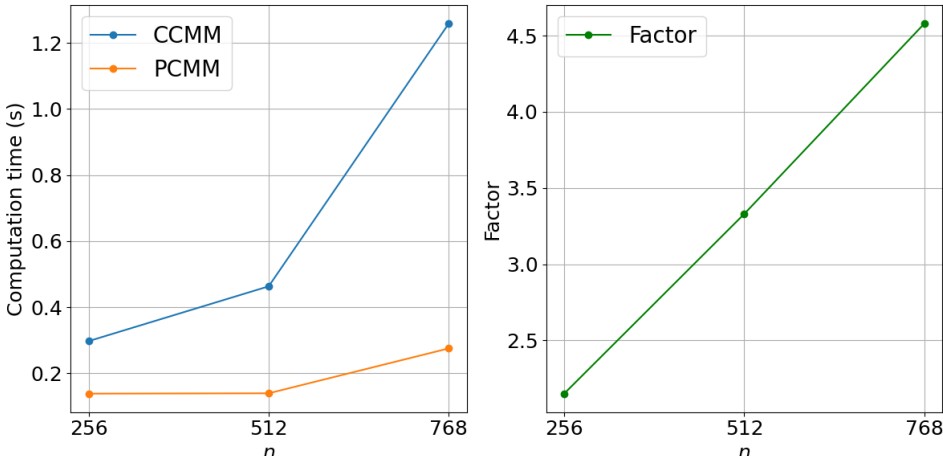

Figure 8: Computation time comparison between PCMM with CCMM for $128 \times n$ matrix by $n \times n$ matrix multiplications according to input dimension $n$. As the input dimension increases, CCMM becomes progressively slower compared to PCMM.

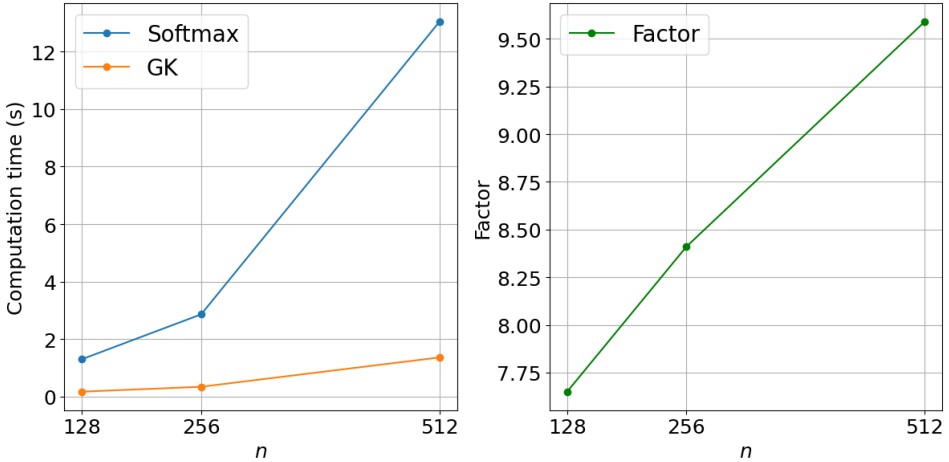

Figure 9: Computation time comparison between Softmax and GK according to input dimension $n$. As the input dimension increases, Softmax becomes progressively slower compared to GK.

## H  CLASSIFICATION ACCURACY AND PRECISION

Here, we present the classification accuracy and average precision of our HE model during inference with fine-tuned weights, comparing the results with evaluation under plaintext. Based on these figures, we can conclude that the inference results under HE are closely aligned with those in plaintext.

Table 16: Classification accuracy and average precision, compared with plaintext inference results. Since STSB is not a classification task, we could not calculate the precision. Acc indicates how closely the class obtained from HE evaluation matches the class predicted in plaintext evaluation. HE inference results are similar to the results in plaintext inference.

|                  | COLA  | RTE    | MRPC   | STSB  | SST2   | QNLI  |
|------------------|-------|--------|--------|-------|--------|-------|
| Acc              | 0.99  | 0.99   | 1.00   | -     | 1.00   | 0.99  |
| Avg prec. (bits) | -9.32 | -13.48 | -10.97 | -8.03 | -11.15 | -9.59 |

