# OpenReview forum: "Encryption-Friendly LLM Architecture"
_ICLR.cc/2025/Conference — ICLR 2025 Poster_

### Official Review · Reviewer_EsSP · 2024-10-30

**Soundness:** 3
**Presentation:** 3
**Contribution:** 2
**Rating:** 5
**Confidence:** 3

**Summary:**

This paper starts from the poor performance problem of homomorphic encryption (HE) in transformer architecture and focuses on optimizing the speed of HE in it. Specifically, the scheme tries to avoid CCMM computations to solve the poor performance brought by full fine-tuning. In addition, this paper aims to address the difficulty of evaluating under HE with Softmax, the core idea of which is to replace Softmax with a Gaussian kernel. Finally, this paper carried out many experiments, and the results show that their scheme is comparable to existing schemes in terms of modeling performance, while at the same time, the computational speed has been significantly improved. The overall narrative of the paper is clear, and has good logic.

**Strengths:**

This paper is oriented to the problem of inefficiency of transformer architecture under HE, although the existing research has produced richer results. The main contribution of this paper is to enhance the speed of transformer architecture under HE, and the authors have carried out many experiments to verify the rationality and advantages of the scheme. I think the experiments in this paper are full, and the advantages of this paper are elaborated in terms of speed and model performance, which speed is emphasized in this paper. The experimental results are thorough and well-analyzed. Overall, this paper seems to incorporate some of the SOTA approaches in the current field and apply them to a widely researched topic.

**Weaknesses:**

1. Insufficient innovation. First, the topic chosen for this paper is a more widely studied one. Second, the solutions in this paper seem to be a direct combination and application of existing advanced schemes, and it is not intuitively obvious in the paper that the authors have improved on existing methods.
2. The description in 2.1 does not seem to be consistent with Figure 1. Furthermore, why does the statement “LLM weights are protected in the strict cryptographic sense (line 149)” hold?
3. Although the paper proposes a privacy-protecting LLM architecture, the security considerations of the model, especially against attacks and model theft, is insufficient.

**Questions:**

1.	One core idea of this paper to solve the difficulty of evaluating Softmax under HE is to replace it with Gaussian kernel, however, this method is very similar to that in the literature “Chen, Yifan, et al. Skyformer: Remodel self-attention with gaussian kernel and nystr\" om method. Advances in Neural Information Processing Systems 34 (2021): 2122-2135”. However, this paper does not describe the difference with this paper or even cite this literature. Is there a difference in the performance of the two? Furthermore, the polynomial approximation method of this paper seems to be very common way.
2.	In Section 3, is the author's approach in solving Bottleneck 1 just an application of some existing methods? If not, please clarify the differences and improvements.
3.	Does the use of LoRA and Gaussian Kernel affect the interpretability of the model? How does the author balance efficiency and explainability?

---

> ### Author Response · Authors · 2024-11-25
> **Individual responce to reviewer EsSP**
>
> We are happy to hear that the reviewer found our work valuable and convincing. We respond to the individual comments in the following.
>
> **First, the topic chosen for this paper is a more widely studied one. Second, the solutions in this paper seem to be a direct combination and application of existing advanced schemes, and it is not intuitively obvious in the paper that the authors have improved on existing methods.**
>
> We ask the reviewer to refer to our common response.
>
> **The description in 2.1 does not seem to be consistent with Figure 1.**
>
> Thank you for raising this point. We revisited Section 2.1 and Figure 1 but could not fully understand the reviewer's concern about consistency. Could the reviewer kindly clarify which aspect in unclear or inconsistent? We would certainly want to address this issue.
>
> **Why does the statement “LLM weights are protected in the strict cryptographic sense (line 149)” hold?**
>
> Our paper states that the LLM weights are **not** protected in the strict cryptographic sense. This is because although the pre-trained LLM resides on the server side, their weights are not encrypted and model theft attacks are not mitigated by our approach.
>
> **Although the paper proposes a privacy-protecting LLM architecture, the security considerations of the model, especially against attacks and model theft, is insufficient.**
>
> We agree with the reviewer, and this is a point we acknowledge in our paper. We emphasize that our focus is on protecting the privacy of the user and protecting the server (model) is a separate and very active thread of research.
>
> **One core idea, Replacing Softmax with Gaussian kernel to solve difficulty of evaluation is very similar to Skyformer [2]. However, this paper does not describe the difference with this paper or even cite this literature. Is there a difference in the performance of the two?**
>
> Thank you for this precise question. We would like to use this response as an opportunity to convey the rationale for using GK.
>
> First and foremost, we clarify that we were aware of Skyformer, but omitted it in discussing the prior work as a simple honest mistake. (We did reference similar prior work such as SOFT [1].) We have incorporated the reference in the updated paper.
>
> The difference between the Skyformer and our contribution is as follows. First, we view Gaussian kernel from a different perspective from Skyformer. On the one hand, the original self-attention is formulated as
>
> $$
> \text{Attention}(Q,K,V)=\text{softmax}\left(\frac{QK^\top}{\sqrt{p}}\right)V=D^{-1}AV
> $$
>
> where $A_{ij}=\exp\left(q_i^\top k_j/\sqrt{p}\right)$ ($q_i$ and $k_j$ are the $i$-th and $j$-th row of the query and the key, respectively) and $D$ is a diagonal matrix with diagonal element $\exp\left(QK^T/\sqrt{p}\right)\cdot \mathbf{1}$. The authors of Skyformer view this as a normalization of $A$, the attention scores, by $D$.
>
> Skyformer modifies Gaussian kernel as
>
> $$
> \text{Kernelized-Attention}(Q,K,V)=\exp\left(-\frac{\\|Q-K\\|^2}{2\sqrt{p}}\right)V=\left(D_Q^{-1/2}A D_K^{-1/2}\right)V
> $$
>
> where $D_Q$ and $Q_K$ are diagonal matrices with $\left(D_Q\right)\_{ii}=\exp\left(\frac{\|q_i\|^2}{\sqrt{p}}\right)$ and $\left(D_K\right)_{ii}=\exp\left(\frac{\|k_i\|^2}{\sqrt{p}}\right)$ where $q_i$ and $k_i$ mean the $i$-th row of the query and key matrices, respectively. From this modification, the authors of Skyformer view Gaussian kernel as another normalization version of the attention scores through $D_Q$ and $D_k$. On the other hand, we interpret Gaussian kernel as an attention scoring mapping `without normalization'. Second, Skyformer applies Nystr\"om method to approximate the kernel to accelerate the calculation. However, we do not use such an approximation. Third, the motivation of Skyformer is to handle long sequence lengths (not a concern for us) while our motivation is to avoid normalization operations, which are difficult to implement under HE.
>
> **In Section 3, is the author's approach in solving Bottleneck 1 just an application of some existing methods? If not, please clarify the differences and improvements.**
>
> We ask the reviewer to refer to our common response.
>
> **Does the use of LoRA and Gaussian Kernel affect the interpretability of the model? How does the author balance efficiency and explainability?**
>
> The interpretability and explainability of deep learning models are very important to using these models more safely. However, this issue is not a focus of our work, especially since we consider the setup where the entire computation is meant to be hidden behind homomorphic encryption.
>
> [1] Jiachen Lu, Jinghan Yao, Junge Zhang, Xiatian Zhu, Hang Xu, Weiguo Gao, Chunjing Xu, Tao Xiang, and Li Zhang. SOFT: Softmax-free transformer with linear complexity. Neural Information
> Processing Systems, 2021.
>
> [2] Yifan Chen, Qi Zeng, Heng Ji, and Yun Yang. Skyformer: Remodel self-attention with gaussian
> kernel and nystr\"{o}m method. Neural Information Processing Systems, 2021.

---

### Official Review · Reviewer_f3Ds · 2024-11-01

**Soundness:** 3
**Presentation:** 4
**Contribution:** 3
**Rating:** 8
**Confidence:** 4

**Summary:**

As LLMs surge, privacy issue becomes important considering government-level regulations. Among methods in privacy-preserving machine learning, homomorphic encryption (HE) can provide cryptographic security by computing over encrypted data directly without extra communication like MPC. However, HE is not efficient to compute matmul or non-poly operations in the transformer-level scale. This work focus on fine-tuning stage with LoRA and softmax variant to create HE-friendly transformer architecture.

**Strengths:**

1. Compared to MPC approaches, n on-interactive property helps HE to be feasible to compute over large-scale LLMs without including considerable communication overhead among computing parties.
2. This work has a great focus on the fine-tuning stage to make LLMs secure for users, which also concentrates on the key components like attention layers in the transformer, and it is also combined with SoTA techniques like LoRA to make the process more efficient.
3. Writing with bottleneck-improvement pattern for LoRA and GK looks good for readers to figure out key ideas.

**Weaknesses:**

1. When you mentioned SoTA LLMs, you should notice that decoder-based models have been proved very powerful in generative tasks. After iteration of the recent few years, BERT series is not as useful and prevalent as decoder models. Hence, the significance to protect BERT-based model is less essential in the current LLMs.
2. Although this work introduces how HE and CKKS work in the secure way, this work does not specify adversary model, such ability of adversaries, type of adversaries (e.g., semi-honest or malicious) and kind of attacks (e.g., member inference attack) this architecture counters with.
3. In the conclusion, this work is too vague on the future work. For example, how cryptographic community develops helps the improvement of this work (e.g., any change on HE). Also, how LLMs itself evolve may change security issue based on this work.

**Questions:**

N/A

---

> ### Author Response · Authors · 2024-11-25
> **Individual responce to reviewer f3Ds**
>
> We are happy to hear that the reviewer found our work valuable, particularly for addressing fine-tuning in the non-interactive setup. We respond to the individual comments in the following.
>
> &nbsp;
>
> **After iteration of the recent few years, BERT series is not as useful and prevalent as decoder models. Hence, the significance to protect BERT-based model is less essential in the current LLMs.**
>
> Thank you for this precise point. We chose the encoder-only model BERT due to the computational constraints. (Decoder-only models require a larger minimum scale to ``make them work'' compared to BERT.) Whether our findings with BERT translate to decoder-only models should be tested in future work, but we are quite optimistic since the architectural forms between the two types of model are quite similar with the only difference being the causal masks in the attention layers (and we can straightforwardly incorporate causal masks into attention layers with Gaussian kernels).
>
> &nbsp;
>
> **This work does not specify adversary model, such ability of adversaries, type of adversaries (e.g., semi-honest or malicious) and kind of attacks (e.g., member inference attack) this architecture counters with.**
>
> Our model is based on the semi-honest security model. The client owns the personal data and sends these with HE encrypted ciphertext to the server. The server conducts fine-tuning and inference under HE. In our model, the client will know only the inference result, and the server only knows the encrypted client data. As a result, the entire security model relies on the semantic security of the underlying CKKS scheme.
>
> We have clarified the adversary model in our revised paper.
>
> &nbsp;
>
> **This work is too vague on the future work. For example, how cryptographic community develops helps the improvement of this work (e.g., any change on HE). Also, how LLMs itself evolve may change security issue based on this work.**
>
> We believe that homomorphic encryption (HE) will play a role in the future use of some language model applications. Our work contributes to this progress by initiating considerations of architecture design specifically tailored for HE. While the landscape of large language model (LLM) research is evolving rapidly, we anticipate that the findings of this work will remain relevant amidst these changes, since we are targeting fundamental components of the transformer architecture.

---

> > ### Comment · Reviewer_f3Ds · 2024-11-30
> >
> > Thanks for your explanation! I agree with you that decoder-only models are too costly and it is probably an interdisciplinary research for both AI and cryptography community. My questions have been clarified. Great work!

---

### Official Review · Reviewer_YQ2M · 2024-11-02

**Soundness:** 4
**Presentation:** 4
**Contribution:** 3
**Rating:** 6
**Confidence:** 3

**Summary:**

This paper presents optimizations to LLM architectures to make them more friendly towards Homomorphic Encryption evaluation. Concretely, first they propose to use LoRA for fine-tuning, where the data used for fine-tuning is given in encrypted form. This is useful to avoid re-training a lot of parameters, since LoRA focuses on updating only a few weights. This minimizes the amount of homomorphic operations w.r.t. plain fine-tuning. They also observe that LoRA is useful for reducing the dimension of homomorphic matrix multiplications. Secondly, they replace Softmax-based with Gaussian Kernel attention. They show that this is a simpler function to evaluate in HE, leading to significant savings with little impact in accuracy.

**Strengths:**

The problem statement is well motivated. Several works are currently exploring evaluation of LLMs under FHE, and the potential applications are also quite compelling. This is an extremely difficult task in terms of achieving viable efficiency, and any method that advances the state-of-the-art in this direction is welcome. The results achieved here show that several optimizations in other domains, that is, LoRA and the use of Gaussian Kernels, turn out to be useful for the evaluation of LLMs in FHE. I am not aware of this observation being made and explored in this depth in other papers (it is worth mentioning https://arxiv.org/pdf/2410.00433, which appeared after the submission deadline).

**Weaknesses:**

I am not particularly impressed by the novelty of this paper. It uses existing FHE tools with existing ML optimizations. This may not be a weakness on its own given the positive results of combining these techniques, but I still think the improvement factors may not be big enough for these techniques to become "enablers" of private LLM applications in practice. Put differently, I am not convinced that the gains here are a significant enough to overcome the blockers that prevent LLMs + FHE from becoming more widely spread.

**Questions:**

For reproducibility, can the authors comment on the source code? Whether they intend to make it public for validation?

Also for reproducibility (and in lack of code), I am interested in understanding better the polynomial approximations used. In Section E the authors talk about penalizing the model in a "pre-training stage". I am not sure I understand how this would work, especially in the context of secure inference (no fine-tuning). What do the authors mean exactly? The model owner retrains the model using this new loss function? Most prior works take a pre-trained model, changing only its non-linearities by polynomial while keeping its weights. What are the authors proposing to do here? Does this require changing the model's weights? Re-training?

---

> ### Author Response · Authors · 2024-11-25
> **Individual responce to reviewer YQ2M**
>
> We are happy to hear that the reviewer found our work well-motivated and the results valuable. We respond to the individual comments in the following.
>
> &nbsp;
>
> **It is worth mentioning https://arxiv.org/pdf/2410.00433, which appeared after the submission deadline.**
>
> Thank you for providing us with the reference to the P3EFT paper. This work focuses on energy consumption, so the focus is different from ours, but there is certainly some overlap in the techniques, so we have included the reference in the updated manuscript.
>
> &nbsp;
>
> **I am not particularly impressed by the novelty of this paper.**
>
> Regarding the novelty of our work, we ask the reviewer to refer to our common response.
>
> &nbsp;
>
> **For reproducibility, can the authors comment on the source code? Whether they intend to make it public for validation?**
>
> Yes, we will provide the source code for the experiments. All of the code for the experiments under plaintext will be provided as PyTorch code. For the experiments under HE we will provide an implementation for LoRA-style CCMM (Appendix D) and polynomial approximations in Python linked with the HE transformer backbone written in C++. Recall that our HE code is not high-level PyTorch code, so it is taking us some time to organize the code into a GitHub repo.
>
> &nbsp;
>
> **Also for reproducibility (and in lack of code), I am interested in understanding better the polynomial approximations used.**
>
> Our code for the polynomial approximations will provide full clarity on the precise details on the approximations we used.
>
> &nbsp;
>
> **In Section E the authors talk about penalizing the model in a "pre-training stage". I am not sure I understand how this would work, especially in the context of secure inference (no fine-tuning). What do the authors mean exactly? The model owner retrains the model using this new loss function?**
>
> We clarify that we pre-train (in plaintext) a new transformer model that is compatible with the Gaussian kernel. Because of the computational constraints of HE, the language model is a manageable size in plaintext, so this pre-training is not a significant computational cost, compared to the cost of inference and fine-tuning under HE.
>
> The range penalizing loss in the following is applied to the last 100 steps of pre-training.
>
> $$
> \mathcal{F}=\left\\{\left(f,L_f\right)\big| \text{The approximation range of } f \text{ is } \left[-L_f, L_f\right]\right\\}
> $$
>
> $$
> L_{\text{penalty}}(X;\theta)= L_{\text{pre}}(X;\theta)+\lambda\\!\\!\\!\sum_{(f,L_f)\in\mathcal{F}}\mathbf{1}\_{\\{\\|X\_f\\|\_{\infty}>L\_f\\}}(X)
> $$
>
>
>
> Here, the loss term multiplied by $\lambda$ is added to $L_{\text{pre}}$ in the last 100 steps of the pre-training. During these 100 steps, the input domains of the non-polynomial functions in $\mathcal{F}$ are narrowed to $\left[-L_f, L_f\right]$ so that it lies in the approximation domains. Other papers such as [1] use this penalizing loss so that the inputs of the non-polynomial functions do not deviate for inference. We found that using this new loss term prevents the input range in the fine-tuning from being too wide.
>
> &nbsp;
>
> [1] Itamar Zimerman, Moran Baruch, Nir Drucker, Gilad Ezov, Omri Soceanu, and Lior Wolf. Converting transformers to polynomial form for secure inference over homomorphic encryption. International Conference on Machine Learning, 2024.

---

> > ### Comment · Reviewer_YQ2M · 2024-11-26
> >
> > Thanks for your response.
> >
> > I see the contribution on identifying LoRA as useful in this setting.
> >
> > I don't think this was addressed in your answer though: you don't intend to make the source code open? I understand the engineering difficulties this direction faces. However, if the source-code is not open I don't think this could pass as a contribution itself. Secondly, I'm not asking for the code only for the polynomial approximations -- which is what your response seemed to imply. Closed code in FHE makes it hard to replicate runtimes by future works.

---

> > > ### Author Response · Authors · 2024-12-02
> > > **We are releasing our code**
> > >
> > > Thank you for the response. We now see that we were not sufficiently clear with our clarification. We meant that we **do** intend to make the code fully open; we were just saying this will take time due to the authors' other personal and work schedules. However, given that the reviewer considers open-sourcing the code an important priority, we hurried up the process. The following anonymous GitHub repo
> > >
> > > https://github.com/Encryption-friendly/Encryption-friendly_LLM_Architecture/tree/main
> > >
> > > provides our code (everything, not just the polynomial approximation stuff) in "version 0". By version 0, we mean that all of the code is provided and can be executed to reproduce all experiments of our paper, but things are not necessarily cleaned up or well-documented. Version 1 will be in a more presentable and user-friendly format, and it will be ready a few weeks down the line.
> > >
> > >
> > > We appreciate and agree with the reviewer's insistence on open and reproducible science. Given that our fully open-source code resolves the main concern of the reviewer, we kindly ask the reviewer to consider increasing the score.

---

### Author Response · Authors · 2024-11-25
**Common response**

We sincerely thank all reviewers for their insightful comments. We are happy to hear that all reviewers found our work to be well-motivated and the writing to be clear.

&nbsp;



At the same time, however, there were some concerns about the simplicity of our approach, particularly the observation that our solutions are direct applications of existing schemes. Indeed, our contribution is not in the novelty of such mechanisms themselves. Rather, our contribution is identifying a novel application of these existing mechanisms.

&nbsp;

- ### **Use of LoRA**

The use of LoRA is not an obvious choice. In the standard plaintext setting, LoRA is used primarily to reduce the memory footprint, but the memory footprint is not a concern of our work under HE. Our contribution is finding a new purpose for LoRA: reduction in computational cost by avoiding large ciphertext-ciphertext matrix multiplications (CCMMs).

&nbsp;

- ### **Use of Gaussian kernel (GK)**
The use of Gaussian kernel (GK) is also not an obvious choice. In the standard plaintext setting, prior works such as SOFT [1] and Skyformer [2], use GK with a low-rank approximation to handle long sequence lengths, but sequence length is not a concern for us. Our contribution is finding a new purpose for GK: an attention layer that is inherently simpler to polynomially approximate compared to the softmax. (Prior work such as HETAL[3] devotes a significant amount of effort to adequately approximate the softmax function, but we circumvent this difficulty by using the GK.)

&nbsp;

Further, we quickly reiterate two points highlighting our contributions.

- This work represents a somewhat significant engineering effort. There is no existing library for implementing ML models using the CKKS scheme, so we had to develop our implementation code from scratch. This includes implementing backpropagation with LoRA, along with their corresponding CKKS-specific computations, in a multi-GPU setup with data parallelism.


- Our work is the first to successfully fine-tune an entire transformer model (as opposed to fine-tuning only the head) under HE. Considering the significant computational constraints imposed by HE, it was not previously evident that fine-tuning under such conditions would be practically feasible. Our work provides the first positive proof of concept in this direction.

&nbsp;

In our view, our work represents meaningful steps of progress made through simple but non-obvious proposals. We kindly ask the AC and the reviewers to take this perspective into consideration.

&nbsp;

[1] Jiachen Lu, Jinghan Yao, Junge Zhang, Xiatian Zhu, Hang Xu, Weiguo Gao, Chunjing Xu, Tao Xiang, and Li Zhang. SOFT: Softmax-free transformer with linear complexity. Neural Information
Processing Systems, 2021.

[2] Yifan Chen, Qi Zeng, Heng Ji, and Yun Yang. Skyformer: Remodel self-attention with gaussian
kernel and nystr\"{o}m method. Neural Information Processing Systems, 2021.

[3] Seewoo Lee, Garam Lee, Jung Woo Kim, Junbum Shin, and Mun-Kyu Lee. HETAL: Efficient
privacy-preserving transfer learning with homomorphic encryption. International Conference on
Machine Learning, 2023b.

---

### Meta-Review · Area_Chair_QKcG · 2024-12-23

**Metareview:**

All reviewers except one (EsSP) argued for accepting the paper. For this reviewer their main conerns were on #1 a lack of novelty, #2 incorrect descriptions, #3 insufficient model security considerations. Specifically for #1 the reviewer argued that (a) the topic is widely studied, (b) the solutions are a direct combination of existing methods, and (c) it is not clear that the authors have improved on existing methods. I disregard (a), as topic popularity alone should not count against a paper so long as (b) and (c) are dealt with. For (b) the authors argue that even though LoRA and the Gaussian kernel have been applied to LLM architectures, they have been used for very different purposes: LoRA to reduce memory, and the Gaussian kernel to handle long sequence lengths. They are using each in different, unintended ways: LoRA to avoid large ciphertext-ciphertext matrix multiplications, and the Gaussian kernel to find a better polynomial approximation, compared to prior work which directly tried to approximate the softmax function. The reviewer had not response to this. I am convinced by the authors’ arguments on both points. It is clear that LoRA would be useful for HE-friendly fine-tuning. And the observations that one need only approximate part of the Gaussian kernel and that this is well-approximated by an easy-to-compute polynomial approximation is brilliant. For (c), the reviewer did not elaborate what baselines should be compared against. In the paper the authors say “Only a few attempts, such as those by Lee et al. (2022b) and HETAL (Lee et al., 2023b), have explored fine-tuning, focusing exclusively on the classification head while leaving other key components like attention layers and feed-forward networks untouched. P3EFT (Li et al., 2024) also addresses fine-tuning of foundation models, but it concentrates more on energy consumption during fine-tuning.” The authors compare against HETAL and P3EFT was released the day of the ICLR deadline so a comparison against this, while I’d like to see it in the future, is out of scope for this submission. Concern #1 is resolved. For #2 the reviewer argued that Section 2.1 was not consistent with Figure 1. The authors revised both of these things but asked for further clarification to understand the inconsistency. The reviewer did not respond. I found no inconsistencies, so concern #2 is resolved. For #3 the reviewer argues that model security is not addressed. The authors agree and stated that their focus is on user security and that model security is separate. I agree and would further argue that one could apply recent developments in model security (e.g., differential privacy for LLMs) to achieve this. Including a new method for model security would create a very confusing conference submission for which most details would have to go into the appendix. This resolves concern #3. Overall the paper demonstrates an impressive new capability: fine-tuning an LLM (beyond just the heads) with HE, which outperforms recent work. The authors have agreed to release their code which will have a big impact on the community.  Given these things, I vote to accept. Authors: you’ve already made improvements to respond to reviewer changes, if you could double check their comments for any recommendation you may have missed on accident that would be great! The only additional request I would like to make is for you to run an experiment where you vary the approximation levels of LoRA and the Gaussian kernel to show the trade-off between computation speed and model performance. I would really recommend highlighting this aspect of your approach as it allows for much more varied use-cases beyond methods without such tunable knobs. After incorporating these changes the paper will make a nice contribution to the conference!

**Additional Comments On Reviewer Discussion:**

Two reviewers responded to the author feedback (YQ2M, with further clarification questions and f3Ds, with a short response). No reviewers engaged in further discussion of the paper. Please see above metareview for more details.

---

### Decision · Program_Chairs · 2025-01-22

Accept (Poster)